# Phospholipids of APOE lipoproteins activate microglia in an isoform-specific manner in preclinical models of Alzheimer's disease

Nicholas F. Fitz[1,4], Kyong Nyon Nam[1,4], Cody M. Wolfe [1,4], Florent Letronne[1], Brittany E. Playso[1], Bistra E. Iordanova [2], Takashi D. Y. Kozai[2], Richard J. Biedrzycki[1], Valerian E. Kagan [1], Yulia Y. Tyurina[1], Xianlin Han [3], Iliya Lefterov[1✉] & Radosveta Koldamova [1✉]

APOE and Trem2 are major genetic risk factors for Alzheimer's disease (AD), but how they affect microglia response to Aβ remains unclear. Here we report an APOE isoform-specific phospholipid signature with correlation between human *APOEε3/3* and *APOEε4/4* AD brain and lipoproteins from astrocyte conditioned media of APOE3 and APOE4 mice. Using pre-clinical AD mouse models, we show that APOE3 lipoproteins, unlike APOE4, induce faster microglial migration towards injected Aβ, facilitate Aβ uptake, and ameliorate Aβ effects on cognition. Bulk and single-cell RNA-seq demonstrate that, compared to APOE4, cortical infusion of APOE3 lipoproteins upregulates a higher proportion of genes linked to an activated microglia response, and this trend is augmented by TREM2 deficiency. In vitro, lack of TREM2 decreases Aβ uptake by APOE4-treated microglia only, suggesting TREM2-APOE interaction. Our study elucidates phenotypic and transcriptional differences in microglial response to Aβ mediated by APOE3 or APOE4 lipoproteins in preclinical models of AD.

[1] Deparment of Environmental and Occupational Health, University of Pittsburgh, Pittsburgh, PA, USA. [2] Department of Bioengineering, University of Pittsburgh, Pittsburgh, PA, USA. [3] Barshop Institute for Longevity and Aging Studies, San Antonio, TX, USA. [4] These authors contributed equally: Nicholas F. Fitz, Kyong Nyon Nam, Cody M. Wolfe. ✉email: iliyal@pitt.edu; radak@pitt.edu

Late-onset Alzheimer's disease (AD) is age-related dementia characterized by amyloid β (Aβ) plaques, neurofibrillary tangles, and cognitive decline[1]. Inheritance of Apolipoprotein (*APOE*) ε4 allele is the major genetic risk factor for late-onset AD, modifying disease risk and progression in an isoform-dependent manner (ε2 < ε3 < ε4, with *ε4* bringing the highest risk)[2,3]. Isoform differences in amino acids at position 112 affects lipid binding properties of APOE and at 158 the receptor-binding affinity to APOE receptors[4]. APOE is a major lipid transporter and the structural differences result in different lipid binding properties. Thus, APOE4 preferentially binds lower density lipoproteins (LDL, VLDL) whereas APOE3 and APOE2 bind high-density lipoproteins (HDL)[5]. In a recent study we applied shotgun lipidomics to measure the major phospholipid classes and their molecular species in brains of AD patients of different *APOE* genotypes[6]. Our data demonstrated that few of these phospholipids were significantly lower in *APOEε4/c* carriers vs. *APOEε2/c* carriers[6]. In the brain, APOE is primarily secreted by astrocytes and, to a lesser extent microglia, in a lipid-poor form[7]. ABCA1 transporter mediates the transfer of phospholipids and cholesterol to lipid-free APOE, thus forming discoidal HDL particles (reviewed in Koldamova et al.[8]). APOE3 and APOE4 lipoproteins isolated from astrocytes conditioned media (referred to as E3 and E4 lipoproteins) are composed of proteins, cholesterol, and phospholipids and deficient of cholesteryl esters[9]. It has been reported that the APOE lipidation state significantly impacts Aβ aggregation[10] and Aβ clearance at the blood–brain barrier (BBB)[11]. APOE4 isoform is associated with impaired Aβ uptake, clearance, and degradation[12], resulting in higher amyloid load in APP transgenic mice[13,14]. Interestingly, a study by Verghese et al. suggested that APOE influences Aβ clearance in vivo not through direct binding but via its interaction with other cell surface receptors[15].

Microglia can have beneficial effects assisting Aβ clearance and by forming a barrier surrounding amyloid plaques but can also be detrimental, causing chronic neuroinflammation (reviewed in Deczkowska et al[16]. and Hansen et al[17].). Recent comprehensive single-cell RNA sequencing (scRNA-seq) analysis in mouse models of neurodegeneration discovered a subset of microglia named disease-associated microglia (DAM)[18,19]. The upregulation of DAM genes is usually accompanied by a decreased expression of homeostatic microglia genes responsible for maintaining the normal microglia function of surveillance. Among DAM genes are genes associated with increased risk for AD, such as *APOE* and *TREM2*. It has been reported that TREM2 can recognize a variety of ligands including APOE, HDL, and LDL, which could affect microglial interaction with Aβ and APOE[20–22]. APOE and TREM2 are essential for the development of microglia barrier around the plaques[23]. Recently, we demonstrated that the lack of TREM2 differentially affects the phenotype and transcriptome of human APOE3- and APOE4-expressing mice[24]. TREM2 deficiency increased plaque growth concurrently to the impairment of microglia barrier, an effect most pronounced at earlier stages of amyloid deposition. Interestingly, lack of *Trem2* significantly decreased plaque-associated APOE protein only in APOE4 but not in APOE3 mice in agreement with gene expression data.

In this study, after performing lipidomic analysis of native E3- and E4-lipoproteins and brain samples from *APOEε3/3* and *APOEε4/4* AD patients, we identified a common phospholipid signature between their major phospholipid classes. We hypothesized that isoform-specific phospholipid composition of E3 and E4 native lipoproteins would elicit distinct phenotypic and transcriptomic response by microglia. We used in vivo two-photon imaging and determined that the infusion of Aβ with E3 lipoproteins induced a more rapid microglial response than with E4 lipoproteins. Behavioral testing following cortical Aβ infusion demonstrated that both native lipoproteins ameliorated Aβ deleterious effects on cognition, with APOE3 exhibiting a higher efficacy than APOE4. To determine how E3 and E4 lipoproteins affect gene expression in microglia in response to Aβ, we performed bulk and scRNA-seq. Collectively, the data showed that the addition of E3 lipoproteins to Aβ upregulated a higher proportion of DAM genes and is associated with more active transcriptomic response than E4. scRNA-seq identified microglia-specific clusters affected by *Trem2* deficiency, suggesting that lack of *Trem2* impairs the transition of microglia from homeostatic to activating state. Compared to E3, E4-expressing microglia showed a reduced Aβ uptake that was additionally aggravated by *Trem2* deficiency. Together, our findings have elucidated unique phenotypic and transcriptional differences in the microglial response to Aβ in the presence of E3 or E4 lipoproteins.

## Results

**Common APOE dependent phospholipid signature between human AD brains and native APOE lipoproteins.** Previously, we have shown a significant APOE isoform-specific difference in phospholipid content in the brains of AD patients, specifically that AD brains from *APOEε2* carriers have a unique lipid profile[6]. In the present study, we compared the phospholipid profile of native APOE lipoproteins isolated from conditioned media of APOE3 and APOE4 primary astrocytes to those of *APOEε4/4*, *APOEε4* carriers, and *APOEε3/3* brain samples from AD patients (Supplementary Table 1). The primary astrocytes were established from mice expressing human APOE3 or APOE4 isoforms[24]. To measure the most abundant nine phospholipid classes in human brain, we re-analysed multi-dimensional mass spectrometry shotgun lipidomics (MDMS-SL) data used in Lefterov et al[6]., and compared it with a new MDMS-SL analysis of APOE lipoproteins. We found that out of nine phospholipid classes, five (PE, PI, PS, SM, and PA) had a statistically higher level in *APOEε3/3* vs. *APOEε4/4* brains while *APOEε4* carrier levels were in the middle (Fig. 1a). Overall, except of PG, APOE-containing lipoproteins had a similar direction of fold change for each class as the human samples (Fig. 1b). In the lipoprotein dataset, we found that PI, PS, SM, and LPC were statistically higher in E3 vs. E4 lipoproteins (for the abbreviation see the legend of Fig. 1). We also found that total phospholipids concentration was higher in CSF of APP/E3 vs. APP/E4 mice (Supplementary Fig. 1k). Lipoproteins derived from E3/Abca1[het] and E4/Abca1[het] astrocytes[13] were used as negative controls and, as expected, demonstrated half of the phospholipid content of their wild-type counterparts (Supplementary Fig. 1). Correlation analysis (Fig. 1c) demonstrated that a very strong correlation existed between all phospholipid species examined, excluding PG. As shown on Fig. 1d, e, 103 lipid species were commonly identified in the human and lipoprotein samples, most of which were affected in APOE isoform-specific manner in both datasets and were higher in E3 samples.

To confirm the results from shotgun lipidomics of APOE native particles, we applied liquid chromatography-mass spectrometry (LC-MS) and measured five of the phospholipid classes reported to comprise the bulk of brain APOE-containing lipoproteins[9], namely: PC, PE, PI, and PS; while CL was used as a control. The tSNE clustering shown on Fig. 1f demonstrated that the phospholipid molecular species from the two APOE isoforms segregated in separate groups, forming distinct clusters. Similarly to the MDMS-SL results, we found that the negatively charged phospholipids (PI, PE, and PS) were significantly higher in E3 than in E4 lipoproteins while CL was unchanged (Fig. 1g, h). Additionally, specific subspecies of PE and PS (PE36:1, PE36:2, PS36:1, and PS40:6) represented a large portion of the

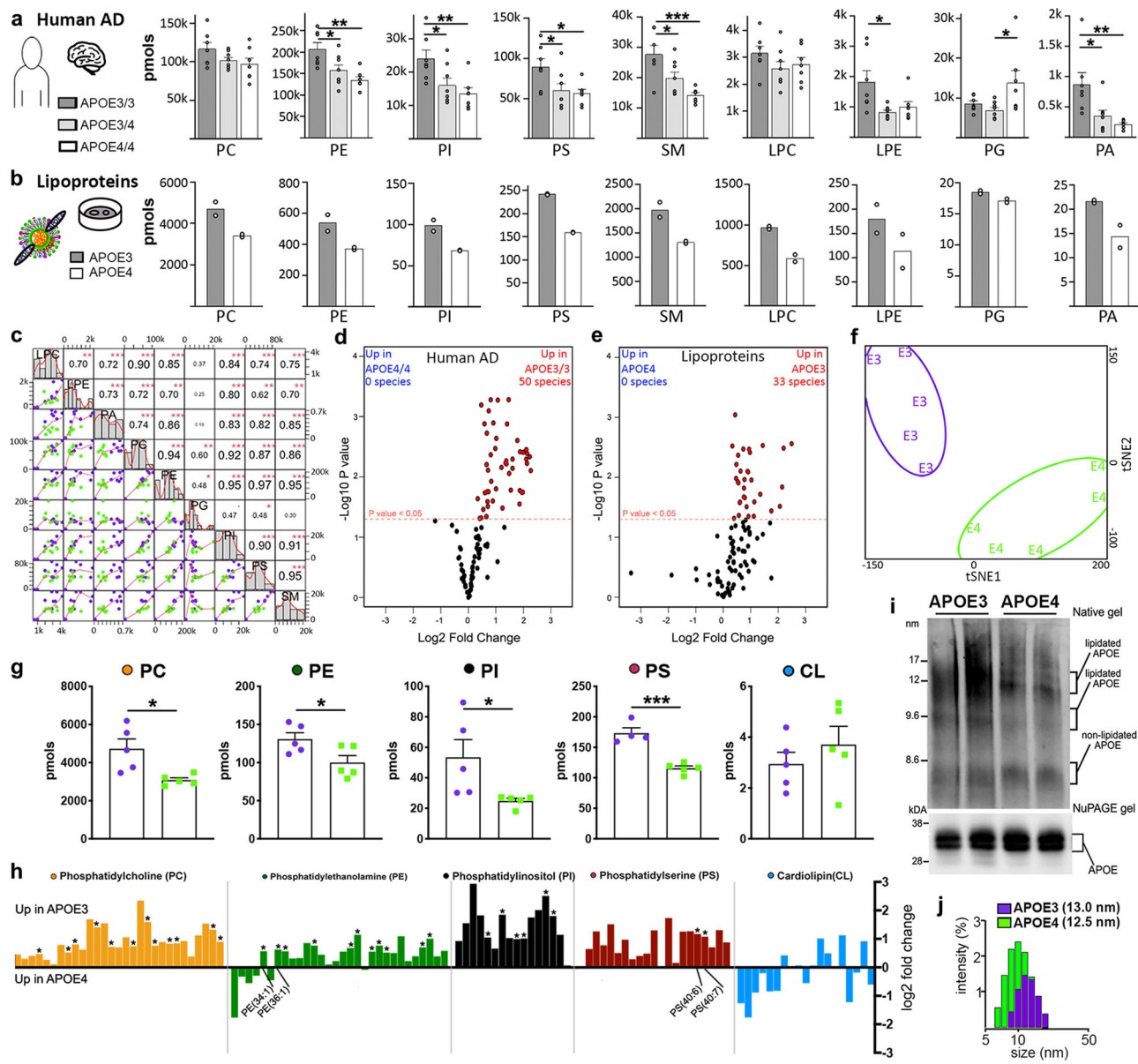

total amount of those classes and were found in significantly less abundance in APOE4 lipoproteins. These specific subspecies may represent potential lipid activation signals which impact receptor binding and could activate signal transduction pathways.

Since the diameter of lipoproteins is commonly used to represent their lipidation[25], we used non-denaturing native gel electrophoresis (Fig. 1i) and dynamic light scattering assay (Fig. 1j) to determine the diameter of APOE native particles. The results demonstrated that the size of E3 native lipoproteins peaked at 13 nm while the size of E4 particles peaked at 12.5 nm (Fig. 1j). The native gels (Fig. 1i) localized APOE within three populations of particles, migrating around 12, 9, and 8 nm. In agreement with a previous study on HDL, the measurements allowed us to conclude that particles with a diameter between 12 and 9 nm corresponded to discoidal α-HDL and were lipidated, while particles with a size of 8 nm corresponded to pre-β-HDL and were lipid poor/non-lipidated[26]. SDS NuPAGE (Fig. 1i, bottom panel) confirmed that an equal amount of APOE protein was used for native gels. Lastly, Supplementary Fig. 1l shows that the cholesterol content of E3 lipoproteins was slightly but significantly increased compared to E4, with no difference in the

amount of free cholesterol or cholesterol ester (Supplementary Fig. 1m).

**APOE lipoproteins rescue Aβ-induced memory impairments.** Prior studies demonstrated that the injection of oligomeric Aβ into brain caused significant memory deficits[27,28]. Here we tested the effect of native E3 and E4 lipoproteins on Aβ oligomerization as before[29]. As visible from Supplementary Fig. 2a–c, Aβ incubation with either E3 or E4 native lipoproteins virtually abolished the formation of oligomers. In fact, we were able to detect Aβ oligomers only when Aβ was incubated alone without E3 or E4. Thus, in terms of preventing Aβ oligomer formation both E3 and E4 native particles seem equally effective at preventing it as determined by our detection methods. Next, to examine if the addition of native APOE lipoproteins to Aβ ameliorates Aβ-induced memory deficits, Aβ ± APOE (referred to as AβE3 or AβE4) were infused into the cortex of young WT mice followed by behavioral testing as shown on Fig. 2a–d. For both tests, Aβ oligomers significantly worsened cognitive performance compared to their corresponding negative control (Fig. 2b, d). In

**Fig. 1 Common phospholipid signature between APOEε4/4 and APOEε3/3 human AD brains and APOE3 and E4 native lipoproteins. a, b** The phospholipid compositions of *APOEε3/3* (*n* = 7), *APOEε3/4* (*n* = 8), and *APOEε4/4* (*n* = 7) brains from AD patients and native APOE3 and APOE4 lipoproteins were determined using MDMS-SL. Brain samples were from the inferior parietal lobe. Native lipoproteins were derived from pooled APOE3 or APOE4 astrocyte conditioned media and two samples of each were measured. Bar charts of nine lipid classes analyzed by Shotgun lipidomics in human (**a**) and native lipoprotein (**b**) datasets. Analysis is by ANOVA followed by Tukey's multiple comparison test. PE, *p* = 0.0256 (E33 vs. E34), *p* = 0.0021 (E3/3 vs. E4/4); PI, *p* = 0.0432 (E33 vs. E34), *p* = 0.0094 (E3/3 vs. E4/4); PS, *p* = 0.0464 (E33 vs. E34), *p* = 0.0274 (E3/3 vs. E4/4); SM, *p* = 0.0357 (E33 vs. E34), *p* = 0.0007 (E3/3 vs. E4/4); LPE, *p* = 0.0139 (E33 vs. E34); PG, *p* = 0.0326 (E3/3 vs. E4/4); PA, *p* = 0.0177 (E33 vs. E34), *p* = 0.0039 (E3/3 vs. E4/4). **c** Correlation matrix of the nine lipid classes analyzed in human and lipoprotein lipidomics data shows correlation scatter plots (bottom-left), histograms of lipid amount (diagonal), and correlation coefficients between lipid classes (top-right). Purple dots denote *APOEε3/3*, green dots denote *APOEε4/4*. Axes, pmol for each lipid species. **d, e** Volcano plots depict all 103 lipid species commonly identified in the human (**d**) and lipoprotein (**e**) datasets. Red denotes species which are significantly upregulated in *APOEε3/3* brains or APOE3 lipoproteins. X axis, log2 fold change and y axis, −log10P value for all lipid species. **e–j** Data for native APOE3 or APOE4 lipoproteins derived from astrocyte conditioned media. **f–h** LC/MS analysis of five Phospholipid classes of APOE3 and E4 lipoproteins. **f** tSNE plot shows unique clustering of all lipids from lipoprotein samples according to APOE isoform: E3 (purple) and E4 (green). **g** Bar charts of major lipid classes of E3 and E4 native lipoproteins comprising all the lipids species shown in panel **h**. Analysis by two-tailed unpaired *t*-test. PC, *p* = 0.0152; PE, *p* = 0.0390; PI, *p* = 0.0417; PS, *p* = 0.0003. **h** Bar chart representing fold change of E3 compared to E4 lipoproteins for individual lipid species from five lipid classes. Significantly impacted species which are found in abundance from the PS and PE classes are labeled. **i** Top: Native gel electrophoresis probed for APOE. The diameter of native markers (in nm) are shown on the left. The migration of lipidated and non-lipidated APOE is shown on the right indicated with brackets. Bottom: SDS-NuPAGE gel followed by Western blot for APOE. **j** The size distribution of E3 (purple) and E4 (green) native lipoproteins was assessed by dynamic light scattering using a Zetasizer. Lipidomic analysis was performed on two independent APOE preparations measured in duplicate. The phospholipid content for each APOE isoform was normalized to APOE protein amount as determined by SDS NuPAGE. Bars represent mean ± SEM. *\*p* < 0.05; \*\*p* < 0.01; \*\*\*p* < 0.001. CL Cardiolipin, LPC lysophosphatidylcholine, LPE lysophosphatidylethanolamine, PA phosphatidic acid, PC phosphatidylcholine, PE phosphatidylethanolamine, PG phosphatidylglycerol, PI phosphatidylinositol, PS phosphatidylserine, and SM sphingomyelin.

contrast, injection of both AβE3 or AβE4 significantly ameliorated the effects of Aβ on cognitive performance. Importantly, the cognitive performance in mice infused with AβE4 remained significantly worse compared to their AβE3 counterparts when tested by novel object recognition (NOR). For both tests, behavior controls showed no significant differences between the groups (Supplementary Fig. 2d–f). We concluded that both E3 and E4 native lipoproteins decreased Aβ aggregation and ameliorated its deleterious effects on cognitive performance, with E3 demonstrating a better efficacy.

To investigate the effect of AβE3 and AβE4 infusions on transcriptome, we injected them into the cortex and performed RNA-seq using sorted microglia and neurons. There were no differentially expressed genes (DEGs) when comparing AβE3 vs. AβE4 neuronal transcriptomes (Fig. 2e). In contrast, there were 1792 DEGs in microglia: 722 upregulated in mice injected with AβE3 and 1072 genes upregulated in AβE4 injected mice (Fig. 2f and Supplementary Data 1). As shown on Fig. 2g, f, upregulated in AβE3 vs. AβE4 microglia were many microglia-specific genes (*Tmem119*, *P2ry13*, and *Trem2*) or genes associated with reactive microglia such as cytokines and cathepsins (Supplementary Data 1). Furthermore, biological processes upregulated by AβE3 injections were more connected to activated microglia phenotype (GO:0002376–immune system process, GO:0031663–lipopolysaccharide-mediated signaling pathway). In contrast, the genes upregulated in AβE4 injected mice and biological process associated with them were less specific to microglia and their activation (Fig. 2h and Supplementary Data 1).

We also examined the morphology of microglia surrounding the injection site using IBA1 staining and Imaris filament module to trace microglial projections (Fig. 2i, j). The heatmap of microglial projections around the infusion site demonstrated that microglia in the AβE3 group had significantly increased branch length and branch points (bar graph insets), and the projections were oriented more toward the infusion site (the area between heat maps) when compared to the AβE4 group. We also performed macrophage-specific F4/80 immunohistochemistry analysis. Figure 2k–m shows an increased F4/80/*Adgre1* immunostaining around the infusion site in mice injected with AβE3

compared to AβE4, which is consistent with the overall reduction in *Adgre1* gene expression seen in AβE4 microglia.

We also examined whether microglial barrier surrounding amyloid plaques in APP/PS1 mice depends on APOE isoform by using mice expressing human APOE3 and APOE4 (referred to as APP/E3 and APP/E4[24]). As shown on Supplementary Fig. 3a, b, linear regression analysis demonstrates a statistically significant relationship between microglia barrier and Aβ plaques in the cortex of APP/E3 mice but not in APP/E4 mice, resulting in less plaques in APP/E3 mice. This suggests that in APP/E3 mice microglia barrier increases around developing plaques, thus restricting their growth. Furthermore, Imaris tracing of microglial processes shows diminished branch number and length in APP/E4 compared to APP/E3 mice (Supplementary Fig. 3c–f), similar to the acute effect after the Aβ infusion in WT mice. Thus, the results from gene expression, microglia morphology, and microglia barrier in APP mice suggest that E3 induce stronger microglial response when compared to E4.

**Microglia interaction with Aβ is affected by APOE isoform.** We first compared the effect of E3 and E4 lipoproteins on Aβ uptake using primary microglia isolated from WT mice. Aβ alone and Aβ preincubated with AβE3[het] lipoproteins were used as control. As shown on Supplementary Fig. 4a–e, labeled Aβ uptake was significantly increased after co-incubation with either of the lipoproteins, but the effect of native E3 was stronger.

To determine if the presence of native E3 and E4 lipoproteins affects microglia function in vivo, we used Cx3cr1[GFP] mice that express EGFP protein in microglia and peripheral mononuclear cells and fluorescently-labeled Aβ. First, we tested the impact of lipoproteins on Aβ uptake of microglia by injecting AβE3 or AβE4 into the cortex and performing flow cytometry at 4 and 24 h post-injection (Fig. 3a–c and Supplementary Fig. 5a–c). The cell suspension was separated on microglia without Aβ (green signal, single+) and microglia that had engulfed Aβ (green + red signal, dual+). As shown on Fig. 3b, c, E3 lipoproteins were more efficient than E4 in facilitating Aβ uptake by microglia at 24 h, but not at 4 h after the injection.

To further examine if APOE native lipoproteins have an impact on the microglial response to infusion of Aβ, we employed

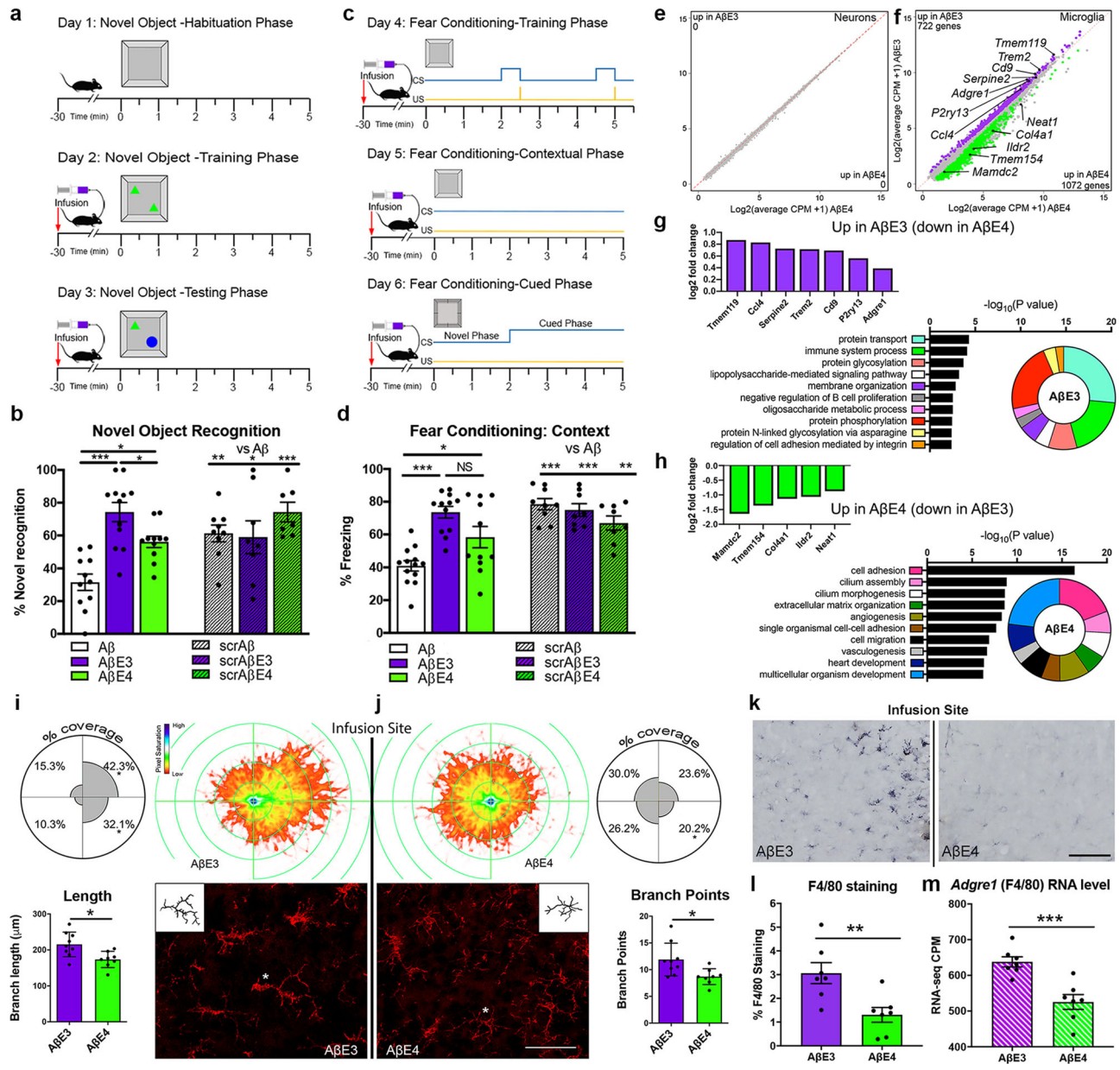

**Fig. 2 Effect of native E3 and E4 lipoproteins on Aβ induced cognitive decline and microglial response. a–d** Aβ preparations after 24 h incubation (Aβ, white; AβE3, purple; and AβE4, green) were infused into the brain of 2-month-old WT mice and cognitive performance examined by novel object recognition (**a**, **b**) and fear conditioning (**c**, **d**). Scrambled Aβ (scrAβ) was used as negative control (scrAβ + Veh, scrAβE3, and scrAβE4; line pattern). All mice were infused 30 min prior to training on days 2 through 6 of behavioral testing as shown on **a** and **c**. Analysis by one-way ANOVA followed by Tukeys multiple comparison test. Aβ vs. AβE3, $p < 0.0001$; Aβ vs. AβE4, $p = 0.0044$; AβE3 vs. AβE4, $p = 0.0351$; Aβ vs. scrAβ, $p = 0.0088$; Aβ vs. scrAβE3, $p = 0.0251$; Aβ vs. scrAβE4, $p = 0.0002$ for novel object recognition. Aβ vs. AβE3, $p < 0.0001$; Aβ vs. AβE4, $p = 0.0388$; Aβ vs. scrAβ, $p < 0.0001$; Aβ vs. scrAβE3, $p < 0.0001$; Aβ vs. scrAβE4, $p = 0.0015$ for fear conditioning. $n = 12$ for Aβ ± E3 or E4, and $n = 8$ for scrAβ controls. The same Aβ preparations were infused into the cortex of another group of WT mice and 24 h later microglia and neurons from one cortex were sorted via MACS and used for RNA-seq. $n = 5$ mice per group. **e**, **f** Scatterplots of neuronal (**e**) and microglial (**f**) gene expression profile from AβE3 vs. AβE4 injected. Genes significantly upregulated at FDR <0.05 in AβE3 (purple), and AβE4 (green). **g**, **h** Bar plots of significantly upregulated genes and GO terms of AβE3 (**g**, purple) and AβE4 (**h**, green) groups. **i**, **j** IBA1 staining around the infusion site of the same AβE3 (**i**, purple) or AβE4 (**j**, green) groups. Imaris filament tracer was used for generation of a microglial projection heatmap around the infusion site. Bottom: representative images of IBA1 staining with Imaris tracing of the labeled (*) microglia as an inset. Scale bar = 50 μm. $n = 3$ mice/group; 86 microglia/AβE3 and 106 microglia/AβE4. Bar graphs show increased branch length and points of microglia around the infusion site of the AβE3 (purple) vs. AβE4 (green) group. Analysis by two-tailed unpaired t-test; $p = 0.0103$ for branch length and $p = 0.0173$ for branch points. **k–m** Representative images of the F4/80 (Adgre1) immunostaining around the infusion site of AβE3 or AβE4 group (**k**). Images are positioned with the infusion site in the middle. Scale bar = 100 μm. **l** Bar graph showing percent coverage of F4/80 staining increased around the infusion site of the AβE3 (purple) vs. AβE4 (green) group. $p = 0.0069$ (**m**) Bar graph showing mRNA level of Adgre1 as determined by RNA-seq (CPM, count per million, line pattern, $p = 0.0007$). Analysis by two-tailed unpaired t-test. $n = 7$ mice per group. Bars represent mean ± SEM. * $p < 0.05$, **$p < 0.01$; ***$p < 0.001$; NS no significance.

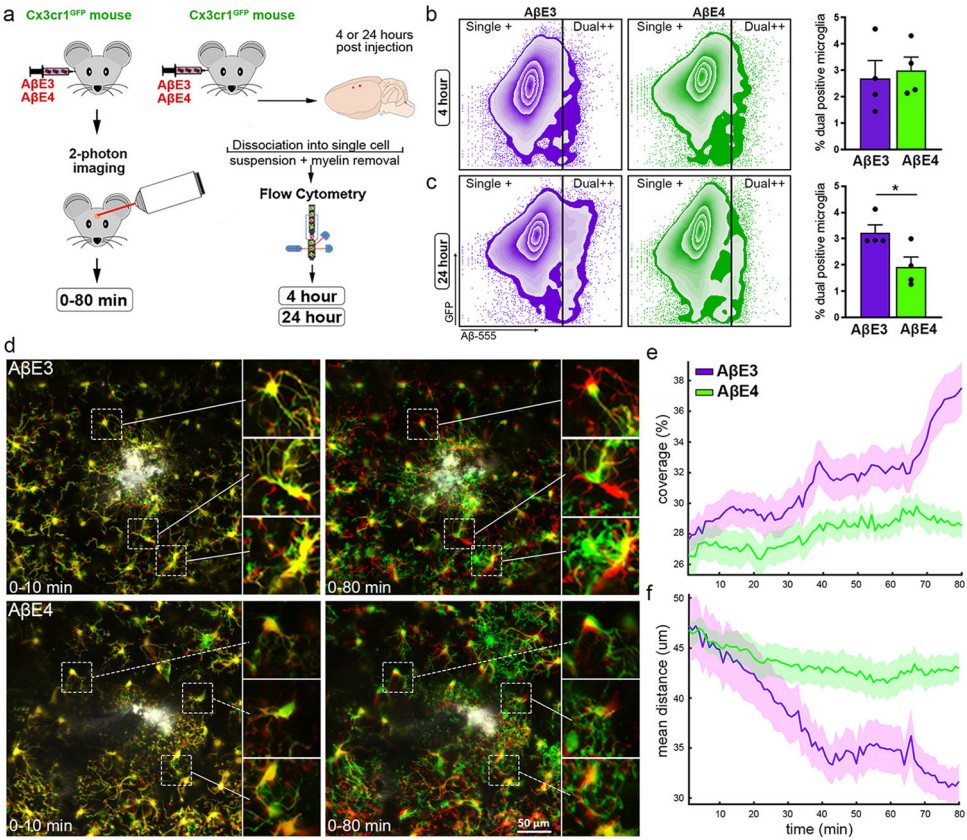

**Fig. 3 Microglial interaction with Aβ is affected by APOE isoform.** Cx3cr1$^{GFP}$ mice were infused with Hi-Lyte Fluor 555-labeled Aβ (Aβ−555) preincubated with native E3 or E4. **a** Experimental design. **b**, **c** Four or 24 h post-injection the cortex surrounding the injection site was dissociated into a single cell suspension and used for flow cytometry (**b**, **c**) and FACS (experiment shown on Fig. 4). Flow cytometry zebra plots represent live microglia cells from injected mice that are GFP$^{high}$/Aβ−555$^{low}$ and are counted as single+ cells or GFP$^{high}$/Aβ−555$^{high}$ cells counted as dual+ cells. Flow cytometry zebra plots from 4 h (**b**) and 24 h (**c**) time points and bar plots correspond to the percentage of dual-positive microglia cells in AβE3 (purple) and AβE4 (green) groups at 4 and 24 h. $p = 0.0372$ by two-tailed unpaired $t$-test. $n = 4$ mice/group. Bars represent mean ± SEM. **d**–**f** Two-photon imaging was used to track microglia movement in real time. **d** Representative fields of view from AβE3 (top row) or AβE4 (bottom row) infused mice. Time series are color-coded with infused Aβ as white; and initial time point in red and later time points in green as indicated on the panel, (0–10 or 0–80 min) with yellow representing the overlap between timepoints. Inserts show details of individual cells demonstrating the extent of migration and directionality of the cellular processes. Scale bar = 50 µm. **e** Percent coverage of infused Aβ by microglial processes computed from channel overlap. **f** Change of mean distance from microglia to infusion site over time for AβE3 (purple) and AβE4 (green) groups. The standard error within each group is plotted as a bounded line with mean as center line. $n = 4$ mice/group.

in vivo two-photon imaging in Cx3cr1$^{GFP}$ mice injected with AβE3 or AβE4 (see Fig. 3a for the design). We injected the same volume and assessed the biodistribution post-injection by acquiring in vivo three-dimensional z-stacks (Supplementary Fig. 5d). For each animal, ZT-stack images were analyzed to quantify the microglial migration towards the infusion site and microglial coverage of the infusion site over time (Fig. 3d–f and Supplementary Fig. 5e, f). The representative images of the time series are color-coded with the initial timepoint (0 min) in red, later timepoints (10 or 80 min) in green and the infused Aβ in white (Fig. 3d and Supplementary Fig. 5e, f). The time series (Fig. 3d, insets) show clear separation of the green and red labels in the AβE3 infusion group (Supplementary Movie 1), indicative of faster migration and better coverage of the infused Aβ compared to AβE4 (Supplementary Movie 2). In contrast, the time series from the mice infused with AβE4 demonstrated a visible overlapping—yellow-colored—representing a slower migration velocity and less microglial Aβ coverage. The percent coverage by the microglial processes over time was computed as the increase of the green channel intensity over the infusion site and shown on Fig. 3e. The change in the distance from microglia to the infusion site is presented on Fig. 3f. Microglia were able to

extend processes towards the AβE3 infusion over the first 40 min. Once the processes reached the AβE3, microglia formed lamellipodia that covered the AβE3 throughout the remainder of the experiment (40–80 min). In contrast, microglia demonstrated decreased capacity to extend processes towards the AβE4 as well as encapsulate AβE4 infusion. The results demonstrate that the infusion of Aβ together with E3 lipoproteins, unlike with E4, induces a more rapid response by microglia to isolate Aβ that suggests a protective mechanism diminishing the spread of Aβ.

**E3 and E4 lipoproteins differentially affect microglia transcriptome in response to Aβ infusion.** Next, we performed RNA-seq on two FACS-sorted microglial populations (dual+ and single+ microglia) from Cx3cr1$^{GFP}$ mice injected with AβE3 or AβE4 (see Fig. 4a for the design). We assume that dual+ cells are in an active stage compared to their single+ counterparts, which are either resting or in transition from homeostatic to active state. The goal was to examine how APOE isoform and the post-injection interval affects this transition. tSNE plots demonstrate that the expression profiles of single+ and dual+ microglia cluster into sharply defined groups for both timepoints (Fig. 4b).

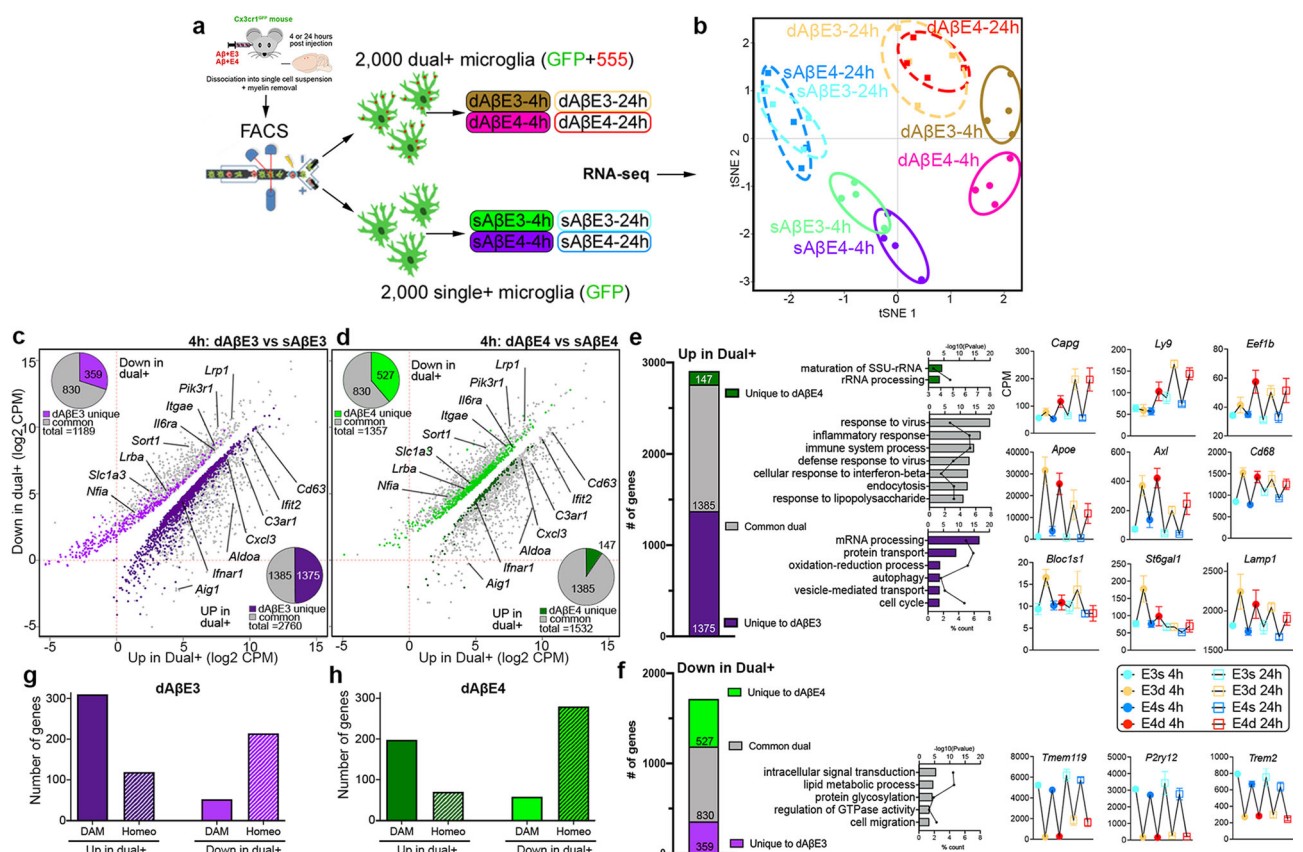

**Fig. 4 AβE3 and AβE4 differentially affect microglia transcriptome.** Cx3cr1GFP mice were infused with AβE3 or AβE4, brain and processed followed by FAC sorting microglia at 4 and 24 h into single+ (referred to as sAβE3 or sAβE4) and dual+ (dAβE3 or dAβE4) populations and used for RNA-seq. **a** Experimental design. **b** tSNE plot shows clustering of dual+ and single+ microglia for four experimental groups. *n* = 4 mice/group **c**, **d** Significantly affected transcripts for dual+ vs. single+ microglia at 4 h time point (FDR <0.05) for AβE3 (purple) and AβE4 (green) groups. **c** DEGs in dAβE3 vs. sAβE3. **d** DEGs in dAβE4 vs. sAβE4. Pi charts in the upper left (down in dual+), and lower right (up in dual+) represent genes common between, or unique to each APOE isoform. **e**, **f** Proportion of genes, GO terms and line patterning associated with common (gray) or unique genes between dAβE3 vs. sAβE3 (purple) and dAβE4 vs. sAβE4 (green). **e** Common and unique upregulated in dual+ vs. single+. **f** Common and unique downregulated in dual+ vs. single+. For both, bar plots indicate the −log10P value for each term, and the associated point in the center of each bar represents the percent of submitted genes found in each GO term. Line-patterning graphs show example of genes from each category. sAβE3 light blue, dAβE3 beige, sAβE4 blue, dAβE4 red; 4 h solid cirles, 24 h open circles. **g**, **h** Bar graphs show the number of DAM (solid) and homeostatic (dot pattern) genes upregulated or downregulated in dual+ for AβE3 (**g**, purple) or AβE4 (**h**, green) injected mice.

Supplementary Fig. 6a, b demonstrates that numerous overlapping DEGs were commonly upregulated for the two timepoints in dual+ vs. single+ microglia. For each single+ and dual+ populations, we identified genes with a similar level of expression at both timepoints (unchanged, gray), a group with higher expression at 4 h that declined in time (upregulated at 4 h, blue), and a third group (red) with higher expression at 24 h (Supplementary Fig. 6c–f and Supplementary Data 2). Unchanged in dual+ cells were genes activated from the start with continuous expression and related to interferon signaling (Ifi204) or lipid transport (Lpl). Early-responding genes upregulated in dual+ at 4 vs. 24 h were associated with inflammatory response, chemokine-mediated signaling, and chemotaxis (Supplementary Fig. 6c, d, blue color and Supplementary Fig. 6g). Genes that were upregulated at the later timepoint in dual+ microglia (late-responding) were associated with protein folding, ribosome biogenesis, and glycolysis, consistent with the increased energy demands of processing the engulfed Aβ (Supplementary Fig. 6c, d, red color and Supplementary Fig. 6g).

As seen from the tSNE plot on Fig. 4b, APOE isoform-specific effect on microglial transcriptomes was more pronounced at 4 h. The examination of DEGs in dual+ vs. single+ microglia in both treatments revealed a higher number of total DEGs, specifically a

higher number of upregulated genes in dual+ AβE3 microglia than in dual+ AβE4 microglia (Fig. 4c, d, 2760, and 1532 respectively, pi charts dark purple and dark green). For both treatments, there were many shared/common DEGs upregulated in dual+ vs. single+ (Fig. 4e, f, gray boxes). In both AβE3 and AβE4 dual+ microglia, there was an overall upregulation of DAM genes (186 genes) and downregulation of homeostatic genes (212 genes). Commonly upregulated in dual+ microglia were DAM genes (Apoe and Axl) while classical homeostatic genes were downregulated (P2ry12 and Tmem119), indicating a profile characteristic for microglia in an active state (Fig. 4e, f). Also commonly upregulated in dual+ were receptors (Clec4 cluster, Marco, Axl, and Gas6), cytokines (nine genes), chemokines (12 genes), and genes related to phagocytosis and lysosomes (Cd68). Surprisingly, Trem2 as well as several other DAM genes (Csf3r, Cebpa, and Ctss) had higher expression in single+ microglia (respectively, downregulated in dual+) in both treatments and at both timepoints. This suggests that Trem2 is more associated with transition than the fully activated state. DAM genes uniquely upregulated for AβE3 were genes associated with the lysosome/ autophagy process (Lamp1 and Bloc1s1), glycosylation (St6gal1), a glycosyltransferase implicated by GWAS with AD[30], cytokines (Ccl6), and receptors (Clec4e and Clec12a). DAM genes uniquely

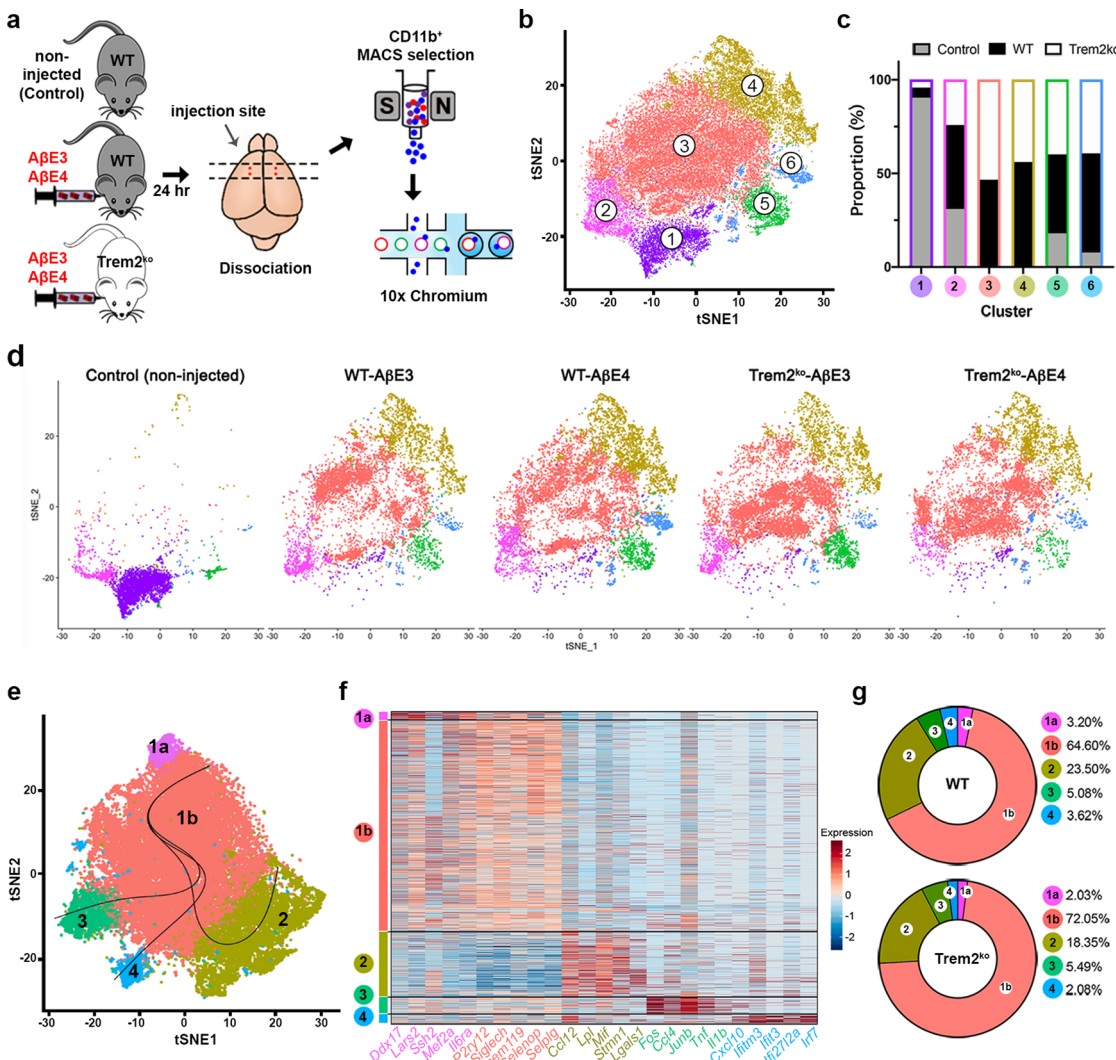

**Fig. 5 scRNA-seq data reveal distinct microglial subpopulations in response to Aβ infusion. a** Schematic workflow of the experimental strategy for scRNA-seq. Microglia were isolated from WT and Trem2ko mice injected with AβE3 or AβE4 using CD11b immunomagnetic beads. Microglia from non-injected WT mice were used as control. Isolated cells were immediately applied to the 10x Chromium platform. n = 1 for Control, n = 2 for WT-AβE3, WT-AβE4, Trem2ko-AβE3, and Trem2ko-AβE4. **b** t-SNE plot shows clusters of 39,898 microglial cells after re-clustering using Seurat. **c** Proportion of cells in each clusters from control, WT, and Trem2ko microglia. **d** t-SNE plots of microglia from each experimental condition. **e** t-SNE plot shows clusters of 36,244 microglial cells after re-clustering of injected samples and inferred trajectories by Slingshot. **f** Heatmap shows the expression of top five genes in each cluster. **g** Proportion of cells in each cluster from WT and Trem2ko microglia.

upregulated in AβE4 were *Capg* (binds PIP2 and affects actin dynamic in macrophages[31]), *Eef2* and *Eef1b2* (regulate translation[19]) as well as *Ly9* (lymphocyte antigen 9[32]). Overall, there was a higher number of DAM genes upregulated in dual+ AβE3 (310) than in dual+ AβE4 (198) (Fig. 4g, h), suggesting that E3 could initiate a stronger response by microglia than E4.

**Single cell sequencing identifies distinct microglial populations in response to Aβ injection.** To test how *Trem2* deficiency affects acute microglial response to AβE3 or AβE4 infused into the cortex of WT and Trem2ko mice, we performed scRNA-seq. Non-injected WT microglia were used as a control for the injection and analyzed in parallel (see Fig. 5a for the design). We collected a total of 60,046 sequenced cells with 2156 median genes expressed and 4988 median unique molecular identifiers (UMIs) per cell. Non-microglial cells were identified using cell type-specific genes and excluded from further analysis (Supplementary Fig. 7a–c); 39,898 microglial cells were re-clustered to catalog

transcriptionally distinct subpopulations (Fig. 5b and Supplementary Data 3). We found that, excluding homeostatic clusters (clusters 1 and 2), four clusters consisted predominantly of microglia from Aβ injected samples, suggesting these four clusters are affected by Aβ injection and not because of experimental artifacts, such as cell isolation (Fig. 5c, d and Supplementary Fig. 8). Thus, we re-clustered 36,244 microglial cells from injected groups, resulting in five clusters and then performed trajectory analysis to identify distinct activation states in response to the infusion (Fig. 5e and Supplementary Fig. 8d, e). We determined three potential trajectories (clusters 2, 3, and 4) from clusters 1a and 1b (Fig. 5e). Sankey plot shown on Supplementary Fig. 8f shows cluster preservation between the two datasets.

To characterize heterogeneity within microglia, we identified cluster-specific genes that are differentially expressed between clusters (Fig. 5f and Supplementary Data 4). As seen in Fig. 5f and Supplementary Fig. 9c, the microglia in clusters 1a and 1b showed increased expression of homeostatic signature genes, such as *P2ry12*, *Tmem119*, and *Siglech*[19,33]. However, cluster 1b showed

an increase from the baseline (cluster 1a) expression of genes related to microglial activation such as *Apoe*, *Cst7*, *Trem2*, and *Tyrobp* and a slight decrease of homeostatic markers (Supplementary Fig. 9c). Thus, we characterized cluster 1b as a transitional state. Interestingly, in WT mice *Trem2* had the highest expression in cluster 1b, suggesting its involvement in the transition from homeostatic to active state. Genes associated with activated microglia were upregulated in clusters 2, 3, and 4 and accompanied by downregulation of homeostatic microglial genes. This suggests that clusters 2, 3, and 4 form in response to infusion of either AβE3 or AβE4 and present the active microglia state. Microglia in cluster 2 showed upregulation of genes related to chemotaxis (*Ccl12*, *Ccl3*, *Ccl4*, *Cxcl16*, *Spp1*, and *Lgals3*), inflammatory cytokine (*Mif*), lipoprotein lipase (*Lpl*), and ribosomal components. Microglia in cluster 3 showed upregulation of pro-inflammatory genes, especially a set of genes important in NF-κB signaling such as *Tnf*, *Il1b*, and *Ptgs2* as well as cytokines (*Ccl3* and *Ccl4*). In cluster 4, genes related to the interferon signaling pathway, including *Ifitm3*, *Ifi27l2a*, *Ifit1*, and *Irf7* as well as *Apoe* were highly expressed within immune system process as the top enriched GO term (Fig. 5f). We observed an increased proportion of cells in cluster 1b and reduced proportion in cluster 2 in *Trem2*-deficient microglia compared to WT (Fig. 5g). These data demonstrate that in our acute model of brain injury, *Trem2* deficiency impairs the transition of microglia from homeostatic to an activated state, in the same way as chronic neurodegeneration[17].

### scRNA-seq identifies activated gene clusters downregulated in Trem2$^{ko}$ mice following Aβ infusion

To identify differential transcripts, we compared microglia from Trem2$^{ko}$ and WT mice and observed a higher number of DEGs in clusters 2, 3, and 4 associated with active microglia (Fig. 6a and Supplementary Data 5). Overall, there was reduced expression of immune response genes in *Trem2*-deficient microglia compared to WT microglia (Fig. 6b–d). In clusters 2, 3, and 4, DAM genes comprised 30–44% of the total number of DEGs that were upregulated in WT vs. Trem2$^{ko}$, and this was accompanied by the same decrease of homeostatic microglia genes (Fig. 6b–d), suggesting that microglia from Trem2$^{ko}$ mice cannot adequately respond to the AβE3/AβE4 infusion challenge. In cluster 2 the top downregulated genes in Trem2$^{ko}$ microglia were *Lpl*, *Spp1*, and *Cd9*; opposed by upregulation of canonical homeostatic genes such as *P2ry12* (Fig. 6b). Clusters 3 (Fig. 6c) and cluster 4 (Fig. 6d) exhibited the largest number of downregulated genes in Trem2$^{ko}$ microglia and also showed the highest mismatch between WT and Trem2$^{ko}$ microglia in terms of homeostatic/DAM gene expression (see the bar graphs in Fig. 6c, d).

We validated the scRNA-seq data using fluorescent in situ hybridization (FISH) for *Tmem119* and *Adgre1* and spatially visualized their expression near the infusion site. We selected *Tmem119* as a canonical microglia-specific gene that was not affected by *Trem2* deficiency, and *Adgre1* (F4/80) was selected because it was significantly downregulated in Trem2$^{ko}$ microglia and was shown to be increased near the infusion site (see Fig. 2k–i). Similarly to the scRNA-seq data, we found a significant reduction in expression of *Adgre1* in Trem2$^{ko}$ mice and no effect on *Tmem119* level (Fig. 6e–g). Thus, FISH performed near the infusion site confirms the result for selected genes in the scRNA-seq datasets.

### Trem2 deficiency and APOE isoform affects microglia response to Aβ

We next examined if the response to AβE3 and AβE4 infusion is APOE isoform- and/or TREM2-dependent. We compared microglia from WT and Trem2$^{ko}$ mice injected with AβE3 or AβE4 within clusters 2, 3, and 4 which correspond to active microglia (Fig. 7a–c and Supplementary Data 6). For all clusters, we observed a higher number of homeostatic genes upregulated in Trem2$^{ko}$ vs. WT microglia injected with AβE4 compared to AβE3-injected groups. Interestingly, the number of DEGs in Trem2$^{ko}$ vs. WT injected with AβE4 showed a larger difference compared to AβE3-injected groups in clusters 2 and 4. In cluster 2, both AβE3- and AβE4-injected Trem2$^{ko}$ groups showed a decreased expression vs. their WT counterparts of genes such as *Cxcl16*, *Lpl*, *Spp1*, *Marco*, and *Ly9*, genes with roles in chemotaxis, endocytosis, and bioenergetic that could affect Aβ uptake. Additional DEGs that were downregulated were several cytokines (*Ccl3* and *Ccl4*), Fc receptors (*Fcgr4*), and *Il1b*. In terms of genes upregulated in Trem2$^{ko}$ vs. WT microglia in cluster 2, AβE4-injected Trem2$^{ko}$ showed a higher number of upregulated classic homeostatic genes such as *P2ry12*, *P2ry13*, and *Tgfb1* compared to AβE3-injected Trem2$^{ko}$ mice (Fig. 7a, AβE4 plot). Similarly, in cluster 4 there was a higher number of upregulated homeostatic genes in Trem2$^{ko}$ vs. WT in AβE4-injected compared to AβE3-injected mice. For cluster 4, in AβE4-injected microglia we also observed a higher number of downregulated genes in Trem2$^{ko}$ associated with interferon signaling (*Ifit2*, *Ifi27l2a*, *Ifi207*, *Ifi204*, and *Axl*) and endocytosis (*Cd14*, *Cxcl16*, *Fth1*, and *Ifitm3*). In contrast, in cluster 4, there were very few DEGs in AβE3-injected Trem2$^{ko}$ vs. WT mice. These data suggest a possible interaction between TREM2 and APOE-isoform that could affect microglial chemotaxis and interferon signaling pathway.

To test how *Trem2* deficiency and APOE native particles affect Aβ uptake, we used primary microglia from WT and Trem2$^{ko}$ mice. As expected, there was a significant difference in uptake between WT-AβE3 and WT-AβE4. Surprisingly, we found that *Trem2* deficiency significantly decreased Aβ uptake only in AβE4- but not in AβE3-treated cells (Fig. 7d, e). Following this, we used primary microglia isolated from mice expressing human E3 and E4 and compared to their Trem2$^{ko}$ counterparts (E3/Trem2$^{ko}$ or E4/Trem2$^{ko}$). Cell were treated with Aβ alone without the addition of E3 and E4 native particle. As seen on Supplementary Fig. 10, both APOE4 isoform and *Trem2* deletion significantly decreased Aβ uptake, and the overall reduction of Aβ uptake by E4/Trem2$^{ko}$ microglia was reduced fivefold compared to E3-expressing microglia. We speculate that the addition of native E3/E4 lipoproteins serves as a ligand to TREM2 or other microglial receptors and facilitates Aβ uptake which is restored to normal with native E3 lipoproteins in Trem2$^{ko}$. Overall, we conclude that *Trem2* deficiency may further exaggerate E4 lipoprotein insufficiency to activate microglia and phagocytize Aβ.

### Discussion

In this study, we identified a common phospholipid signature between major phospholipid classes of native E3 and E4 lipoproteins and brain samples from *APOEε3/3* and *APOEε4/4* AD patients. Our results support the hypothesis that native E3 lipoproteins trigger a more effective microglial response to Aβ than E4 in ameliorating Aβ deleterious effect. We posit that the divergence in phospholipid content of E3 and E4 native lipoproteins affects Aβ binding to microglia receptors in an isoform-specific manner and subsequently has a differential impact on microglia transcriptome (Fig. 8). Consequently, E3 induces a faster microglial response than E4 characterized by rapid process extension allowing Aβ to be isolated and engulfed. Lastly, behavioral testing following Aβ injections demonstrated that E3 native lipoproteins exhibited a better protection than E4 in diminishing Aβ effects on memory.

The quantitative assessment of the phospholipid composition of E3 and E4 native lipoproteins and brain samples from

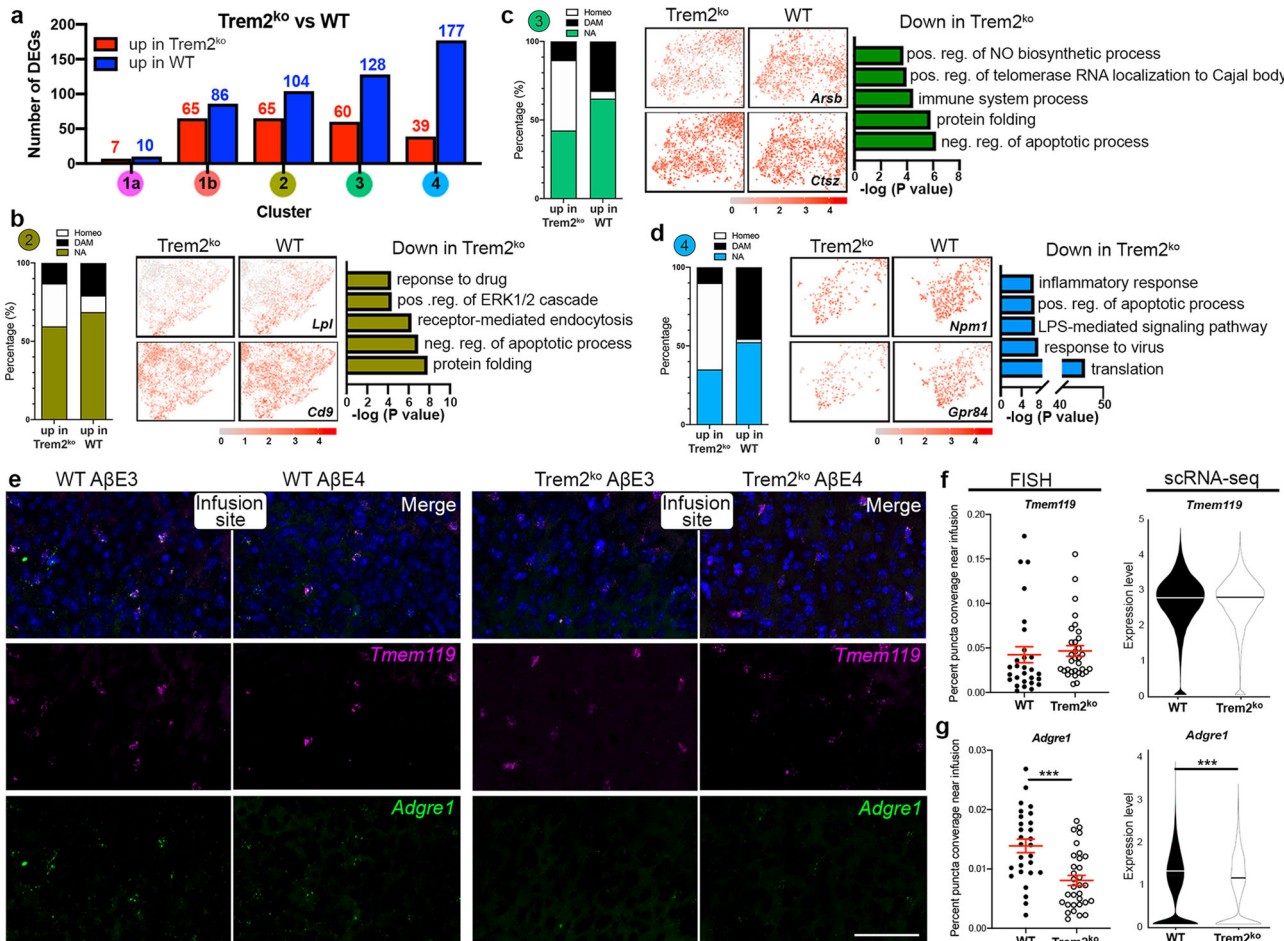

**Fig. 6 scRNA-seq identifies activated gene clusters downregulated in Trem2ko mice following AβE3 or AβE4 infusions. a** The number of DEGs in Trem2ko compared to WT in each cluster (Bonferroni-corrected p value <0.05 and natural logFC >0.1). **b–d** The proportion of homeostatic (Homeo) or DAM genes among DEGs in Trem2ko or WT groups in clusters 2 (**b**), 3 (**c**), and 4 (**d**). Feature plots (Left) showing representative genes and bar plot (Right) showing top five GO terms that are significantly downregulated in Trem2ko compared to WT microglia in the three infusion responsive clusters. n = 4 per group (WT and Trem2ko). **e–g** The expression of Tmem119 and Adgre1 was spatially visualized at 200 μm perimeter from the injection site. **e** Representative images of FISH analyses of microglial gene expression near the Aβ infusion site. (Tmem119 – pink, Adgre1 – Green, Nuclei – Blue). Scale bar = 50 μm. **f, g** Scatter dot plots showing the area of puncta coverage adjacent to the infusion site along with violin plots of the expression of the same gene in the scRNA-seq dataset for Tmem119 (**f**), and Adgre1 (**g**). n = 8 mice per group (equal males and females), with four infusion sites analyzed per animal, red lines represent mean ± SEM, *** p < 0.001 (p = 0.0001) using two-tailed unpaired t-test.

APOEε3/3, APOEε4 carriers, and APOEε4/4 AD patients demonstrated a significant APOE isoform-specific difference in phospholipid profiles. Interestingly, our analysis demonstrates a very strong correlation of the phospholipid profile of human and native lipoprotein samples (Fig. 1c). It should be noted that the human brain samples include lipids that are structural part of cellular membranes as well as lipoproteins in interstitial fluid. Regardless that the source of the phospholipids was not entirely the same in human brain parenchyma and native lipoproteins isolated from ACM, it was remarkable to discover such a high level of similarity in the direction of fold change between the phospholipids of human samples and native lipoproteins that is APOE isoform-specific. As shown on Fig. 1a, several of the phospholipid classes (PE, PI, PS, SM, and PA) demonstrate an apparent APOEε4 allele dose-response effect in the order APOEε3/3 > εE3/4 > εE4/4.

As the major lipid carrier in the brain APOE lipoproteins play an important role in the chaperoning of cholesterol and phospholipids between cells and serving as a ligand for multiple immune receptors such as TREM2. It has been shown that TREM2 signaling is activated by phospholipids such as PI and

PC[34] and TREM2 binds APOE as well as LDL and HDL[22]. Our lipidomic results demonstrate that four major phospholipid classes including neutral (PI and PC) and negatively charged (PE and PS) are significantly reduced in E4 lipoproteins when compared to E3, but not cardiolipin. Sensing of lipids is important for microglia activation and affects functions such as chemotaxis and phagocytosis (reviewed in Chausse et al[35].). Increasing evidence suggests that brain phospholipid levels decrease with aging as well as during AD progression[36], making the initial deficit we have identified in E4 particles even more detrimental with the progression of the pathology. The reduced lipidation of E4 compared to E3 native particles may impact Aβ aggregation which in turn could affect cognitive performance and microglia activation state[37]. Numerous studies demonstrated APOE binds to Aβ with high affinity and most agree that there is an isoform-specific effect on Aβ aggregation E4 > E3 > E2[38,39]. Furthermore, studies have suggested that E3 forms more stable complex with Aβ than E4[39] however in the present study we did not examine the stability of Aβ-APOE complex. Our data demonstrated that both E3 and E4 native lipoproteins precluded the formation of Aβ oligomers (Supplementary Fig. 2a, b). However, we cannot exclude a

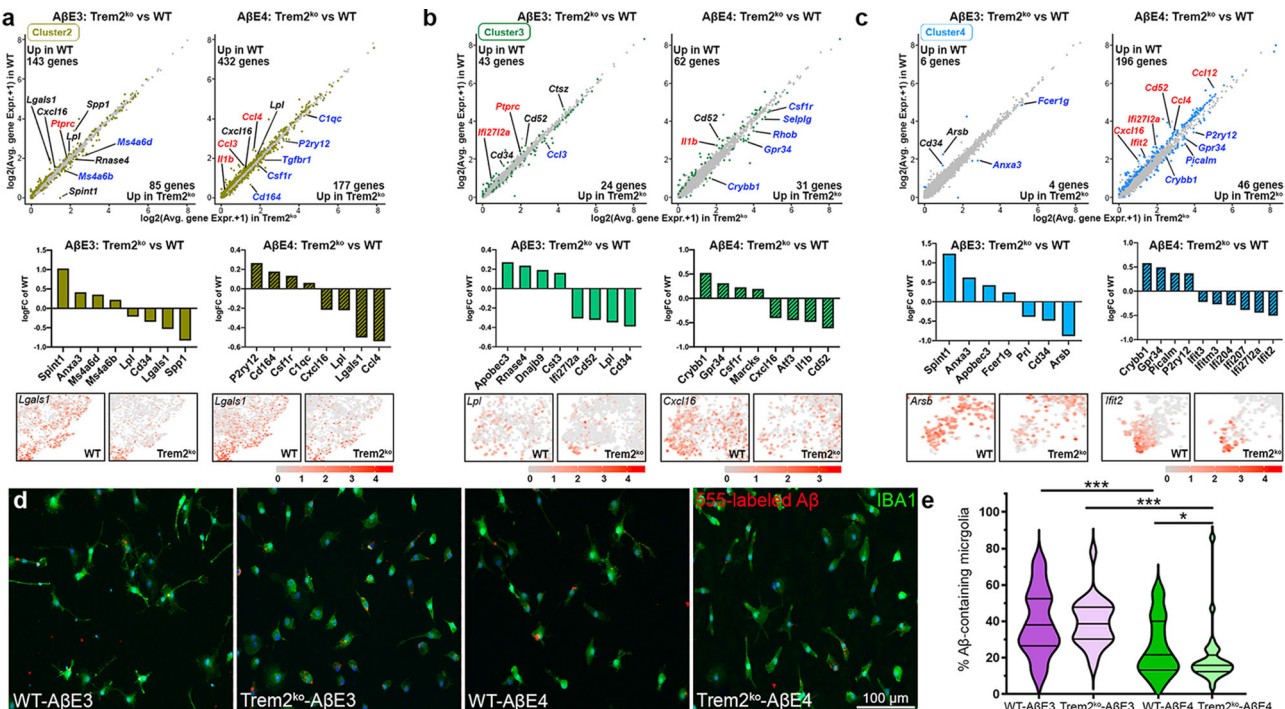

**Fig. 7 Trem2 deficiency and APOE isoform affect microglia response to Aβ. a–c** Scatter plots, bar plots, and feature plots showing the differentially expressed genes in microglia from WT and Trem2ko mice injected with AβE4 or AβE3 complexes for cluster 2 (**a**), cluster 3 (**b**), and cluster 4 (**c**). Differentially expressend genes at $p < 0.01$ are colored in each cluster color. The adjusted $P$ value was performed using Bonferroni correction. Genes labeled with black color indicate commonly up- or down-regulated genes in AβE3 or AβE4 injected mice. Genes labeled with red indicate uniquely upregulated genes in WT-AβE3 or WT-AβE4 and genes labeled blue is uniquely upregulated genes in Trem2ko-AβE3 or Trem2ko-AβE4. $n = 2$ mice per group (WT-AβE3, WT-AβE4, Trem2ko-AβE3, and Trem2ko-AβE4). **d**, **e** Microglia established from WT and Trem2ko mice were treated with and 555-labeled Aβ for 1 h. **d** Representative images of microglia from WT and Trem2ko mice incubated with AβE4 or AβE3 labeled with IBA1 in green and 555-labeled Aβ in red. **e** Bar plot showing the average percentage of Aβ-containing microglia for each group (WT-AβE3 purple, WT-AβE4, green, Trem2ko-AβE3 light purple, and Trem2ko-AβE4 light green). $n = 3$ independent cultures at least in triplicate. Analysis by one-way ANOVA followed by Tukey's multiple comparison test. WT-AβE3 vs. WT-AβE4, $p = 0.0008$; Trem2ko-AβE3 vs. Trem2ko-AβE4, $p < 0.0001$; WT-AβE4 vs. Trem2ko-AβE4, $p = 0.0485$. Violin plots represent kernel densities for each dataset showing median (middle line) and 75% (top) and 25% percentile (bottom). $*p < 0.05$; $***p < 0.001$.

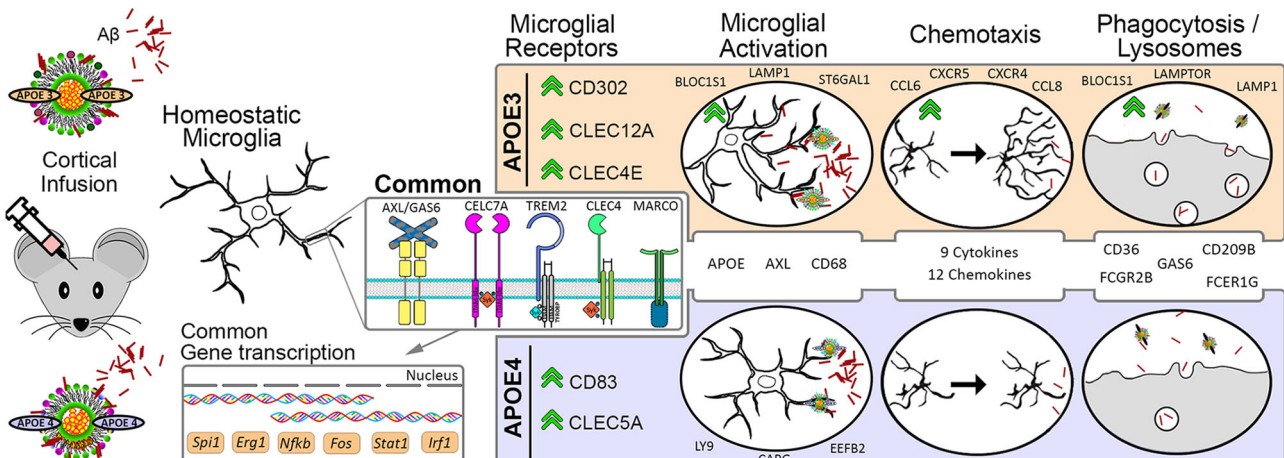

**Fig. 8 Illustration of the APOE isoform-specific phospholipid composition effects on microglia transcriptome and phenotype in response to Aβ.** Shotgun and LC-MS lipidomic analysis of APOE3 and APOE4 astrocyte conditioned media identified isoform-specific differences in phospholipid composition of APOE-containing lipoproteins. E3 or E4 lipoproteins co-infused with Aβ differentially impacted Aβ binding to microglia receptors (AXL, different CLECs clusters, TREM2, CD83, and others) this in turn affects genes expression of transcription factors (*Spi1*, *Stat1*, *Nfkb*, and *Irf1*) resulting in microglia activation. Active microglia demonstrate characteristic transcriptional profile, increased secretion of cytokines and chemokines thus facilitating chemotaxis and phagocytosis. Green arrows indicate an upregulation of that gene or category compared to the other APOE isoform.

possible presence of occasional Aβ oligomers and potentially an isoform-specific effect on Aβ oligomerization that could be detected using quantitative methods with higher resolution. In confirmation of such possibility is the fact that E3 was more effective than E4 in rescuing cognitive deficits caused by Aβ oligomers (Fig. 2a–d). In terms of effects on behavior, we propose that microglia in contact with APOE3 native lipoproteins are more protective by restricting or eliminating the harmful effects of Aβ oligomers on the neural cells and circuits involved in cognitive function.

In terms of APOE effect on microglia, we found that E3 native lipoproteins are more effective than E4 in triggering microglia response to Aβ injections. The transcriptomic profile of microglia isolated from AβE3 injected mice have a higher enrichment of gene expression networks associated with innate immune response compared to microglia from AβE4 injected. To address the transcriptional changes in phagocytic and non-phagocytic microglia we compared dual+ vs. single+ at 4- and 24-h post-injection. Transcriptionally, the dual+ populations are very different from single+ at both 4- and 24-h post-injection, representing an activated state with upregulation of DAM genes and downregulation of homeostatic genes. We further subdivided dual+ population into early-responding genes (4 h; *Ccl9*, *Ifit1*, and *Ctsb*), genes with continuous expression (*Ifi204*, *Lpl*, and *Il10*), or late-responding genes (24 h; *Adgre1*, *Mif*, and *Ccl2*). By separating of microglia on dual+ and single+ we conclude that the APOE- isoform-specific effects are more pronounced at earlier points of the injury that may reflect involvement in the initial recognition of Aβ + APOE complexes by microglia. Furthermore, in Cx3cr1 mice (Fig. 4), AβE3 injections were associated with more active transcriptomic profile comprised by differential increase of DAM compared to AβE4 injections. Recently, Ngueyn et al[40]. using single nuclei RNA-seq of human AD brain identified microglia subpopulation annotated as "amyloid-responsive microglia" (ARM), which was enriched in *APOEε3/3* patients and relatively depleted in *APOEε4* carriers. Ngueyen et al. propose ARM as primed to elicit an activated immune response and note the reduced ARM response in *APOEε4* carriers[40]. We found a number of overlapping genes between our dual+ cells (Fig. 4) and injury responsive microglia from Hammond et al[33]., which contain genes associated with interferon signaling. We found that the number of overlapped genes is higher in dual+ microglia from AβE3 at 4 h than in dual+ AβE4, suggesting E3 induces the change of the microglial phenotype from homeostatic state to interferon responsive microglia better than E4. We also compared our common dual+ signature (Fig. 4) to the list of genes identified by Krassmann et al[41]. as associated to microglia phagocytic function. We identified a similar pattern of overlapping genes such as increased expression of *Apoe*, *Axl*, and decreased expression of *P2ry12* and *Tmem119* that were commonly affected in AβE3 and AβE4 injected mice. We speculate that there is a commonality between the effect of E3 and E4 native lipoproteins on phagocytic transcriptomic profile.

The identified transcriptional changes are accompanied by phenotypic changes in microglial morphology exhibited by increased process orientation towards the infusion site in AβE3 vs. AβE4 injected mice and an increase in microglia which colocalized with Aβ. We confirmed this using in vivo two-photon imaging, demonstrating that within the first hour of infusion microglia exposed to AβE3 extended processes toward Aβ faster than microglia exposed AβE4. Thus, in brains injected with AβE3, microglia can cover and envelop Aβ with better efficiency than microglia from AβE4-injected brains (Fig. 3). Finally, in vivo and in vitro experiments show that E3-native particles are more effective than E4 at promoting Aβ uptake (Figs. 3, 7 and Supplementary Fig. 4). Furthermore, as shown on Supplementary

Fig. 3, APP mice expressing human APOE3 have a higher microglia coverage compared to their APOE4 expressing counterparts and this is accompanied by different morphology. We posit that differences in microenvironment around amyloid plaques, e.g., the composition of APOE-containing lipoproteins, may affect microglia response to amyloid aggregation in plaque form. Our interpretation of the findings from acute Aβ injections and chronic amyloid deposition is that E3 better mobilizes microglia to form a protective barrier surrounding the neuronal damage caused by the Aβ injection and restricts its spread by increasing Aβ uptake.

Recently, we reported that microglia-plaque coverage of small plaques is more efficient at restricting plaque-growth than of larger plaques[24], and this effect is modulated by both APOE isoform as well as TREM2 status. This emphasizes the importance of the early response by microglia to amyloid and implicates the phenotypic response seen in the first 24 h in the AβE3 mice as critical to the long-term protection from amyloid buildup and eventual neuronal damage. Studies addressing the microglia phenotypic response in TREM2-deficient mice have shown impaired ligand binding, reduced cell clustering to amyloid, reduced plaque compaction, and an impaired activation profile[19,21,24,34,42]. Using scRNA-seq we identify clusters 2, 3 and 4 as infusion-responsive clusters characterized by downregulation of homeostatic microglial genes and upregulation of DAM genes as well as other genes associated with active microglia phenotype[19,33,41,43] (Fig. 6). Interestingly, *Trem2*-deficient microglia showed impairment of the transitioning toward the activation state (cluster 1b on Fig. 5) and overall higher expression of homeostatic microglial genes in infusion-responsive clusters (Fig. 6), indicating a defective response to acute Aβ injection. Finally, across all the activated clusters we find larger differences of gene expression in AβE4-injected mice depending on the presence of TREM2 compared to AβE3-injected groups. In clusters 2 and 4, whereas both AβE3- and AβE4-injected Trem2[ko] groups showed a decrease in expression of genes associated with activation vs. their WT counterparts, AβE4-injected Trem2[ko] microglia showed a higher number of upregulated classic homeostatic genes compared to AβE3-injected Trem2[ko] mice. Interestingly, the number of DEGs in Trem2[ko] vs. WT injected with AβE4 showed a larger difference compared to AβE3-injected groups in cluster 4. For this cluster, AβE4-injected microglia showed a higher number of downregulated genes in Trem2[ko] microglia, primarily genes associated with interferon signaling. In vitro uptake experiments, also surprisingly, demonstrated that *Trem2* deficiency significantly decreased Aβ uptake only in AβE4- but not in AβE3-treated microglia. These data suggest a possible interaction between TREM2 and APOE-isoform that could affect microglial chemotaxis and interferon signaling pathways.

Collectively, we found that in the presence of Aβ, E3 lipoproteins are more effective than E4 lipoproteins at inducing rapid transcriptional and phenotypic response by microglia, that can counteract Aβ harmful effects.

## Methods

**Experimental models.** All animal experiments were performed in accordance with the NIH Guide for Care and Use of Animals and approved by the University of Pittsburgh Institutional Animal Care and Use Committee. The following mouse lines were bred to generated experimental mice: Wild-type C57BL/6 J (WT; JAX), human APOE3 and APOE4 targeted replacement mice (E3 and E4; Taconic), *Trem2*[em2ADiuj]/J mice (Trem2[ko]; JAX), and 129P2(Cg)-Cx3cr1[tm1Litt]/J (Cx3cr1[GFP]; JAX). Human APOE3 and APOE4 targeted replacement mice were bred to Abca1[tm1Jdm]/J (Abca1[ko]; JAX) to generate APOE3/Abca1[het] and APOE4/Abca1[het] (E3[het] and E4[het]). Likewise, human APOE3 and APOE4 targeted replacement mice were bred to *Trem2*[em2ADiuj]/J mice (Trem2[ko]; JAX) to generate APOE3/Trem2[ko] and APOE4/Trem2[ko] (E3/Trem2[ko] and E4/Trem2[ko]). Furthermore, APP/PS1dE9 mice expressing mutant familial variants of human amyloid precursor protein (APP; JAX) with Swedish mutation, and human presenilin 1

carrying the exon-9-deleted variant (PSEN1dE9) were crossed to human APOE3 and APOE4 targeted replacement mice. Genotypes were confirmed with primers on Supplementary Table 2. All experimental mice were on the C57BL/6 genetic background, kept on a 12 h light-dark cycle with ad libitum access to food and water and randomly assigned to an experimental group. All reagents were purchased from Fisher Scientific unless documented otherwise. Methoxy-X34 (X34) 1,4-bis (3-carboxy-4-hydroxyphenylethenyl)-benzene, was provided by William E Klunk, MD, PhD, (University of Pittsburgh). For in vivo labeling of dense plaques, we used Methoxy-X04 (X04) 1,4-bis (4'-hydroxystyryl)−2-methoxybenzene, synthesized in the Klunk lab and used according to our previous publication[24].

**AD brain samples**. Human samples (Supplementary Table 1) were provided by the University of Pittsburgh Alzheimer's Disease Research Center (ADRC) brain bank and the Sanders-Brown Center on Aging at the University of Kentucky. The samples were collected according to the protocols approved by the University of Pittsburgh Institutional Review Board and Committee for Oversight of Research Involving the Dead (CORID, No 731). Braak staging was performed on Bielschowsky-stained sections and APOE allelic polymorphism previously determined by a PCR-based assay. Gray matter samples of APOEε3/3, APOEε3/4, and APOEε4/4 genotypes from the right inferior parietal lobule were dissected and used for MDMS-SL. Age and postmortem intervals (PMI) matching was confirmed by t-test.

**Native APOE generation and characterization**. Cultures of primary astrocytes were established from 1-day-old E3, E4, E3het, and E4het targeted replacement pups[44,45]. The cortices and hippocampi were mechanically dissociated using a sterile pasteur pipette and cultivated in DMEM/F12 medium supplemented with 10% bovine growth serum, L-glutamine and antibiotics. Cells were cultured on poly-D-lysine (100 μg/mL) coated T75 Costar flasks (Corning). Confluent primary astrocytes were incubated with de-lipidated serum treatment medium (Neurobasal medium with antibiotics and glutamine as above minus serum) for 48 h. Conditioned medium was collected immediately centrifuged, washed, and filtered through 0.22 μm filter and concentrated using Amicon® Ultra Centrifugal Filters (10 kD cutoff) to retain relatively large particles (between 8–12 nm diameter). Native APOE lipoproteins were washed three times with cold PBS using Amicon® Ultra Centrifugal Filters and their concentration measured by quantitative Western Blotting (NUPAGE) using commercial human APOE (Meridian Life Science) as a standard. Native APOE particles were resolved on Novex™ 4–20% Tris-Glycine gels using an Amersham™ HMW calibration kit as native ladder (GE Healthcare). The size of APOE lipoprotein particles were measured by dynamic light scattering with Zetasizer Nano Z (Malvern). APOE lipoprotein total cholesterol, free cholesterol, and cholesterol ester content was assessed with the Cholesterol Quantitation Kit (Sigma) according to the manufacturer's instructions. Total phospholipid content of mouse CSF was assessed with the Phospholipid Assay Kit (Sigma) according to the manufacturer's instructions.

**Multi-dimensional mass spectrometry shotgun lipidomics (MDMS-SL)**. The MDMS-SL assay[46–48] was performed to determine differences in the lipid composition of the native E3 and E4 particles isolated from primary astrocyte cultures and human AD brain samples described above. APOE particles isolated from E3het and E4het primary astrocytes, exhibiting diminished APOE lipidation states, were used as negative controls. Quantitative analysis was performed on a triple-quadrupole mass spectrometer (Thermo Fisher Scientific) equipped with an automated nanospray apparatus NanoMate and Xcalibur system. Internal standards for the quantification of individual molecular species of the major lipid classes were added to each sample at the start of the extraction procedure. Lipid extraction was performed by the methyl-tert-butyl ether (MTBE) method with resuspension in chloroform/methanol (1:1 v/v) solution with nitrogen flush. Identification and quantification of all reported lipid molecular species scans from the mass spectrometer were automatically acquired with a customized sequence subroutine operated under Xcalibur software (v4.3, ThermoFisher)[47,49]. The resulting MDMS-SL data were analyzed and visualized in R (v3.6.0) and GraphPad Prism (v8.3.1). PC phosphatidylcholine, PI phosphatidylinositol, PE phosphatidylethanolamine, PS phosphatidylserine, SM sphingomyelin, LPC lysophosphatidylcholine, LPE lysophosphatidylethanolamine, PG phosphatidylglycerol, and PA phosphatidic acid.

**Liquid chromatography-mass spectrometry (LCMS) lipidomics**. Phospholipids were extracted and separated as previously performed[50]. MS analysis of phospholipids was performed on a Q-Exactive hybrid-quadrupole-orbitrap mass spectrometer (Thermo Fisher Scientific) using a normal phase column (Luna 3 μm Silica 100 Å, 150 × 2.0 mm, (Phenomenex)) at a flow rate of 0.2 mL/min on a Dionex Ultimate 3000 HPLC system (maintained at 35 ºC). The analysis was performed using A and B gradient solvents containing 10 mM ammonium acetate and 0.5% triethylamine (A— propanol:hexane:water (285:215:5, v/v/v) and B— propanol:hexane:water (285:215:40, v/v/v)). The column was eluted for 0.5 min isocratically at 25% B, then from 0.5 to 6.5 min with a linear gradient from 25 to 40% solvent B, from 6.5–25 min using a linear gradient of 40–55% solvent B, from 25–38 min with a linear gradient of 55–70% solvent B, from 38–48 min using a

linear gradient of 70–100% solvent B, then isocratically from 48–55 min at 100% solvent B followed by a return to initial conditions from 55–70 min from 100 to 25% B. The column was then equilibrated at 25% B for an additional 5 min. Analysis was performed in negative ion mode at a resolution of 140,000 for the full MS scan in a data-dependent mode. The scan range for MS analysis was 400–1800 m/z with a maximum injection time of 128 ms using 1 microscan. An isolation window of 1.0 Da was set for the MS and MS2 scans. Capillary spray voltage was set at 3.5 kV and capillary temperature was 320 ºC. The S-lens Rf level was set to 60. The phospholipid content for each APOE isoform was normalized to the amount of APOE protein as determined by SDS NuPAGE. The analysis of the resulting LCMS data was performed and visualized in R (v3.6.0) and GraphPad Prism (v8.3.1). See lipid class abbreviations in prior section.

**Aβ oligomer preparation**. Aβ42 peptide (American Peptide Company), HiLyte™ Fluor 488-Aβ42, or HiLyte™ Fluor 555-Aβ42 (AnaSpec) were used for all Aβ oligomerizations and injections[27]. Under a fume hood, 0.1 mg Aβ peptide was dissolved in ice cold 1,1,1,3,3,3-Hexafluoro-2-Propanol (HFIP, Fluka) then vortex for a few seconds. The solution was dried with a gentle stream of nitrogen to obtain a peptide film at the bottom of the vial. Prior to use, the film was resuspended in anhydrous dimethyl sulfoxide (DMSO) to form a 5 mM solution, sonicated in a water bath for 10 min and diluted in sterile phosphate-buffered saline (PBS) to a final concentration of 100 μM. HiLyte™ Fluor 488- or 555-Aβ or Aβ42 peptide and E3 or E4 native particles were combined, vortexed, held under oligomer forming conditions (room temperature) for 24 h at a final concentration of 50 μM Aβ and 5 μM APOE and stored at −20 ºC until use (abbreviated AβE3 or AβE4). The same molar concentration (10:1) of scrambled Aβ (AnaSpec) was dissolved in vehicle, combined with isolated E3 or E4 particles, held under oligomer forming conditions and stored at −20 ºC until use (abbreviated scrAβE3 or scrAβE4). scrAβE3 or scrAβE4 was utilized as negative controls throughout the subsequent experiments. Furthermore, Aβ42 peptide and scrambled Aβ was incubated with PBS vehicle as negative control for APOE particles.

**Aβ oligomer characterization**

*Western blot*. Aβ oligomers were examined using western blotting, proteins were resolved on 4–12% Bis-Tris gels (Invitrogen) and transferred onto nitrocellulose membranes. These membranes were probed with 6E10 antibody (Biolegend, 1:2000) and anti-APOE antibody (Millipore, 1:4000) with immunoreactive signals visualized using enhanced chemiluminescence.

*Electron microscopy*. For electron microscopy, 5 μL of each sample was placed on a freshly glow-discharged carbon-coated grid, adsorbed for 2 min and excess solution was blotted using filter paper[27]. The grid was washed with deionized water before staining with 5 μL of freshly filtered uranyl acetate solution (1%, w/v) for 15 s. Excess stain was blotted and the grid was allowed to air-dry. Grids were imaged on a Tecnai T12 microscope (FEI) operating at 120 kV and ×30,000 magnification and equipped with an UltraScan 1000 CCD camera (Gatan) with post column magnification of 1.4x.

**Guide cannula implantation**. To examine the cognitive effects in WT male mice (3 month), Aβ oligomers co-incubated with native APOE lipid particles were infused directly into the brain through implanted guide cannulas[27]. Following anesthesia with isoflurane, the head was shaven and sterilized with two separate povidone-iodine-alcohol washes. A 50% mixture of bupivacaine and lidocaine was applied to the surgical site and ophthalmic ointment applied to the eyes. The head was leveled in a stereotaxic frame and an incision made exposing the dorsal aspect of the skull. Two holes were drilled into the skull (coordinates: AP = −2.46 mm, L = ± 1.50 mm) and 26-gauge guide cannulas (Plastics One) were lowered to a depth of 1.0 mm. Cannulas were fixed to the skull with acrylic dental cement attached to two bone anchoring screws and the surgical opening sutured closed. After suturing, animals were administered buprenorphine and sterile saline, body temperature was maintained until sterna recumbence and animals were allowed to recover for 8 days prior to the start of behavioral testing.

**Behavioral testing**. All stages of behavioral testing were performed at the same time of the day (during the light phase), ensuring 24 h between each phase of testing. Fear conditioning was started 24 h after the completion of NOR. Prior to behavioral testing, mice were placed into individual containers, taken to the behavioral testing room and handled for 3 min for three consecutive days to reduce anxiety. Thirty minutes before a training stage, animals received an infusion of Aβ oligomer co-incubated with either E3 or E4 (Fig. 2a–d). Scrambled Aβ co-incubated with either E3 or E4 was used as a negative control. Mice were randomly assigned to either the AβE3, AβE4, scrAβE3, or scrAβE4 group. The dummy cannulas were removed and infusion cannulas, attached to microsyringe pump by polyethylene tubing, were placed in the guide cannula. Aβ oligomer (final volume of 1 μL per hemisphere) was infused over 1 min, the cannulas were left in place for 1 min to allow for diffusion of the sample and finally dummy cannulas replaced. Each animal received five infusions of Aβ oligomer starting on day 2 of NOR until the completion of behavioral testing (day 6). Between behavioral trials, the paradigms were cleaned with 70% ethanol to eliminate any olfactory cues. Performance

was recorded and scored using ANY-maze software (Stoelting Co.) during all phases of testing.

*Novel object recognition.* NOR was performed over three consecutive days[24]. On day 1, habituation phase, each animal was allowed to freely explore an open arena (40 cm × 40 cm × 30 cm white plastic box) for two 5 min trials with a 5 min intertrial interval. On day 2, familiarization phase, each animal was returned to the arena containing two identical objects (tower of LEGO® bricks) located in opposite diagonal corners for two 5 min trials separated with a 5 min intertrial interval. On day 3, testing phase, the animal was returned to the arena with two objects in the same positions as previously, but one object was replaced with a novel object (metal bolt and nut of similar size). Mice were allowed to explore the two objects for 10 min. The exploration of both objects was defined as the mouse sniffing or interacting while facing an object within 3 cm. Mice were consistently placed into the middle of the arena facing the posterior wall to prevent any object preference. The percent exploration, an indicator of recognition memory, was determined by dividing the time exploring the novel object by the total time exploring both objects. Locomotor activity was assessed by measuring the total distance traveled in the open field during day 1 of testing.

*Contextual and cued fear conditioning.* Contextual and cued fear conditioning (CCFC) was performed over three consecutive days[24]. On day 1, training phase, mice were placed in a conditioning chamber (Stoelting Co.) for 5.5 min. The first 2 min were silent, allowing the mouse to acclimate to the chamber; this was followed by a 30 s tone (2800 Hz; Intensity 85 dB, conditioned stimulus (CS)) ending in a 2 s foot shock (0.7 mA, unconditioned stimulus (US)) through the floor of the conditioning chamber. The process was repeated one more time and ended with 30 s of re-acclimation. On day 2, contextual phase, mice were placed in the same conditioning chamber for 5 min with no tone or shock administered, to measure contextual fear conditioning. On day 3, the gray walls of the chamber were replaced with black and white striped walls to introduce a novel environment for assessing cued fear conditioning. Mice were placed in the conditioning chamber for 5 min. After the first 2 min of silence, the tone was administered for 3 min, to measure cued fear conditioning. Freezing time was defined as the absence of movement except for respiration and calculated as percent freezing of the total time in the chamber during each phase of testing.

*Cortical infusion.* WT male mice (3 month) received cortical infusions of unlabeled AβE3 or AβE4. In a separate cohort of Cx3cr1[GFP], WT, and Trem2[ko] male mice (3 month), HiLyte™ Fluor 555-labeled Aβ combined with native E3 and E4 particles was infused into the cortex. Mice were randomly assigned to either the scrambled Aβ, Aβ alone, AβE3, or AβE4 group. For the bilateral infusion, a 28-gauge infusion cannula (Plastics One) was connected to a 10 μL glass syringe (Hamilton) with vinyl tubing and placed in a micro syringe pump. Following anesthesia with isoflurane, the head was shaven and sterilized with two separate povidone-iodine-alcohol washes. A 50% mixture of bupivacaine and lidocaine was applied to the surgical site and ophthalmic ointment applied to the eyes. The head was leveled in a stereotaxic frame and an incision made, exposing the dorsal aspect of the skull. Four holes were drilled into the skull (coordinates: AP = −1 and −2.5 mm, L = ± 2.0 mm) and the infusion cannula was lowered into the cortex (DV = −1.0 mm). At each infusion site, 2 μL of Aβ preparation was infused at a rate of 0.5 μL/min and the cannula remained in place for 4 min following the infusion. Following the infusions, the surgical opening was sutured closed, animals were administered buprenorphine and sterile saline and placed on a heating pad until fully recovered.

*Animal tissue processing.* Four or 24 h following the cortical infusion, mice were perfused and tissues collected. Mice were anesthetized with Avertin (250 mg/kg of body weight, i.p.) and transcardially perfused with 25 mL of cold 0.1 M PBS, pH 7.4. For Cx3cr1[GFP] and Trem2[ko] mice used for FACS, flow cytometry and single cell RNA-seq the infusion sites in both hemispheres were excised for cellular isolation as described below. For WT mice, one hemisphere was drop fixed in 4% phosphate-buffered paraformaldehyde at 4 °C for 48 h before storage in 30% sucrose. The other hemisphere was used for cellular isolation as described below. Fixed hemibrains were mounted in O.C.T. and cut in the coronal plane at 30 μm sections using a frozen cryotome (Thermo Scientific) and stored in glycol-based cryoprotectant at −20 °C until histological staining. Prior to staining, sections containing the infusion site were selected under a dissection microscope (Olympus). In a separate cohort of APP/E3 and APP/E4 mice (14 month) CSF was collected prior to perfusion. Once anesthetized a small incision was made exposing the foramen magnum, a 28 g 1/2″ needle was used to puncture a hole at the base of the skull and a capillary tube was applied for CSF withdraw.

*Immunohistochemistry.* To characterize changes in activated microglia in the vicinity of the infusion site, a series of six brain sections from each animal was immunostained with anti-IBA1 antibody (WAKO). Free-floating sections were washed, followed by antigen retrieval in sodium citrate buffer at 80 °C for 60 min, blocked in Normal Donkey Serum (Jackson Lab, 5%) for 1 h, and finally incubated in IBA1 antibody (1:1000) overnight at 4 °C. Sections were washed and transferred

into secondary donkey anti-rabbit Alexa 594 antibody (Invitrogen, 1:250) for 1 h, before being washed, mounted on superfrost plus slides and coverslipped. Fluorescent confocal images of the infusion site were taken using a Nikon A1 confocal microscope at 60x magnification with 1.0 μm step size. Four sequential images were captured flanking either side of the infusion site. Analysis was performed as in Marsh et al[51]. with modifications. The FilamentTracer module (Imaris, version 7.1.1, Bitplane) was utilized to trace processes of cortical IBA1-positive microglia and determine process length and number of branch points as indicators of microglial morphology complexity. Using the 3D mapped filament tracings, we oriented all microglia so their central axis was positioned facing the infusion site and generated a heatmap depicting the location of microglia from an experimental group utilizing the ImageJ - heatmap from image stack plugin (v1.53e, National Institutes of Health). The number of overlapping microglial processes increases the pixel saturation metric. The percent of all processes which occupy space in each of the four quadrants was used to determine the percent coverage of that quadrant.

A second series of six brain sections around the infusion site was used for F4/80 immunostaining. First, we quenched endogenous peroxidases with 0.3% hydrogen peroxide, tissues were blocked in 3% normal goat serum (Vector), then blocked for endogenous avidin and biotin. Sections were incubated in F4/80 antibody (Abcam, 1:500) overnight at 4 °C. Sections were washed and transferred into secondary biotinylated anti-rat antibody (Vector, 1:250) for 90 min before being washed and subsequently developed using the Vector ABC kit and DAB substrate kit (Vector). Sections were mounted onto superfrost plus slides and coverslipped. Bright-field images were taken using a Nikon Eclipse 90i microscope at 20x magnification encompassing the infusion site. Image intensity threshold was established to detect the F4/80 staining compared to background using NIS Elements software (Nikon Instruments Inc.) and values were represented as the area of staining normalized to total image area or percentage of area covered.

In a separate cohort of APP/E3 and APP/E4 mice (6 month) free floating section were stained for DAPI and IBA1 as above and six confocal images of the cortex were captured (Nikon A1; 40x, 1.0 μm step size). The morphology of IBA1 positive microglia was determined by seed point labeling of cell bodies and dendrites as described above utilizing Imaris. An adjacent series of brain tissue was used for Thioflavin S (ThioS) staining. Sections were mounted onto slides, washed in PBS, and stained with 0.02% ThioS (Sigma, USA) in PBS for 10 min. Next, sections were differentiated in 50% ethanol for 2 min. After a final wash in PBS, slides were coverslipped. Fluorescent images were taken using a Nikon Eclipse 90i microscope at 4X magnification. To quantify plaque pathology, two separate regions of interest (ROI) were drawn around the cortex and hippocampus for each section and an image intensity threshold was established to detect the stained plaques compared to the background using NIS Elements software (Nikon Instruments Inc., USA). ThioS staining values were represented as the area of staining normalized to ROI area or percentage of area covered by ThioS stain. Microglia process length was correlated to percent ThioS staining with nonparametric correlation analysis.

*In vivo plaque labeling.* X04 (10 mg/kg) was administered intraperitoneally to APP/E3 and APP/E4 mice (3–5 month) and sacrificed 30 days later for in vivo labeling of dense core amyloid plaques in brain[24]. Tissue was processed and free-floating sections were immunostained with anti-IBA1 as above and labeled with goat anti-rabbit Dylight 488 (Di-1488 Vector Labs, 1:250) as described above. Stained tissues were imaged on an Olympus FV1000 confocal microscope at 60x (1.5 μm step size); with plaques selected at least 50 μm from the tissue edge and at least 50 μm away from other plaques, with an even representation of all plaque sizes and composition across groups. Images were loaded into the Imaris (v9.3.1) environment and voxels less than 500 intensity were removed to reduce background noise. Surfaces were then generated for both channels and coverage determined using the "surface-surface contact area" XTension. Sum of colocalized voxels divided by the total surface area of the plaque generated the percent microglia coverage. An adjacent series of brain tissue was used for Thioflavin S (ThioS) staining and analysis as described above. Percent microglia plaque coverage was correlated to percent ThioS staining with simple linear regression.

*Magnetic-activated cell sorting (MACS).* For WT mice infused with unlabeled Aβ, we isolated microglial and neuronal cellular populations utilizing MACS column-based protocols according to the manufacturer's instructions (Miltenyi Biotec). First, the tissue was dissociated utilizing the Neural Tissue Dissociation kit (Miltenyi Biotec), a gentle two-step enzymatic dissociation (papain and trypsin) that yields a high number of viable cells. Briefly, the cortical tissue was enzymatically and mechanically lysed at 37 °C for 35 min to obtain a cell suspension. Then the suspension was passed through a 40 μm cell strainer to remove debris and ensure a single-cell suspension. After a 10 min centrifugation at 300x*g*, the pellet was gently resuspended with ice-cold PBS + 0.5% BSA buffer and incubate for 15 min at 4 °C with Myelin Removal Beads II (Miltenyi Biotec). After washing with PBS + 0.5% BSA and 10 min centrifugation at 300x*g*, the pellet was resuspended with PBS + 0.5% BSA. The single-cell suspension was applied to a column placed on a magnetic stand (LS column and magnets from Miltenyi Biotec) to deplete the myelin fragments by magnetic separation and washed with PBS + 0.5% BSA. Finally, the cells were centrifuged for 5 min at 300 g and the pellet was resuspended with 1 mL of PBS + 0.5% BSA. After obtaining a myelin-free single-cell suspension,

the microglia were labeled with anti-CD11b immunomagnetic beads and isolated using magnetic columns (Miltenyi Biotec). To isolate the neurons, the cell suspension depleted of microglia cells was incubated with a cocktail of antibodies from the Neuron Isolation Kit (Miltenyi Biotec). Non-neuronal cells like astrocytes, oligodendrocytes, endothelial cells, or fibroblasts were magnetically labeled and depleted through the magnetic columns; therefore, neurons were isolated by negative selection. To confirm the purity of isolated microglia, we performed RT-qPCR with *P2ry12* and *Tmem119*, microglia-specific markers (Supplementary Table 2). To further ensure microglia and neuron purity when using MACS, following sequencing and alignment, all genes with low expression (average raw read count <30.9) were removed from further analysis.

**Fluorescence-activated cell sorting (FACS) and flow cytometry**. For the Cx3cr1$^{GFP}$ mice infused with HiLyte™ Fluor 555-labeled Aβ, microglia were isolated utilizing FACS. After removing the cerebellum, subcortical area and olfactory bulbs, cortical tissue within 1 mm of either side of the infusion site was processed into a single-cell suspension using the Neural Tissue Dissociation kit followed by the Myelin Removal Beads II (Miltenyi Biotec) as described above. For WT mice, after the myelin removal steps the pellet were resuspended with 100 μl of PBS and the microglia cells were labeled with 5 μl of anti-mouse/human CD11b antibody (clone M1/70) conjugated with Alexa Fluor® 647 (BioLegend, 1:100) for 20 min on ice. All cells were then washed and prepared with 1 mL of PBS + 0.5% BSA for FACS sorting.

A bio-contained BD FACSAria™ III sorter (BD Biosciences) was used to sort microglia cells for the two experimental populations. First, live cells were separated from the debris according to their forward scatter and side scatter properties and a second gate was used on individual cells only. The GFP fluorescence was detected with a 525 nm filter (488 nm laser) for the microglia isolated from the Cx3cr1$^{GFP}$ mice, the CD11b/APC fluorescence was collected with a 668 nm filter (647 nm laser) for the WT mice and the 555-labeled Aβ fluorescence was detected with a 613 nm filter (555 nm laser). About 2000 microglia with high GFP signal but low 555-labeled Aβ signal, were considered unexposed to Aβ and termed single-positive cells (single+). About 2000 microglia cells with a high signal for both labels (GFP and 555-labeled Aβ) were considered exposed to Aβ and termed dual-positive cells (dual+). The single+ and dual+ microglial populations were sorted into 1.5 mL tubes containing 350 μL of RLT lysis buffer from the RNeasy kit (Qiagen) and 3.5 μL of 2-mercaptoethanol. After the FACS sorting, the bio-contained BD FACSAria™ III sorter (BD Attovision™ software, BD biosciences) was used with the same gates to count and estimate the percentages of the total microglia population and single+ and dual+ microglia population in a total of 1,000,000 live cells.

**RNA isolation and RNA-sequencing**. RNA was isolated from the MACS-isolated microglia and neurons and FACS single+ and dual+ microglia following the manufacturer's instructions for the RNeasy Mini kit and RNeasy Micro kit (Qiagen), respectively. After RNA isolation, concentration was measured with a Qubit 3.0 Fluorometer (Thermofisher) before quality assessment with the 2100 Bioanalyzer instrument (Agilent Technologies). Sequencing libraries for MACS-isolated cells were generated using mRNA Library Prep Reagent set (Illumina) according to the manufacturer's instructions[52]. The total RNA from sorted cells was fragmented and converted into cDNA using the SMART-Seq® v4 Ultra Low Input RNA Kit for Sequencing (Takara Bio). Briefly, a minimum of 10 pg of purified total RNA was used to perform a first-strand cDNA synthesis in a PCR clean workstation. After amplification by LD PCR and purification using the Agencourt AMPure XP beads (Beckman Coulter), 1 μL of the amplified cDNA was used for validation using the Agilent 2100 Bioanalyzer on a High Sensitivity DNA chip. One nanogram of the cDNA was used with the Nextera DNA Library Preparation Kit (Illumina) to prepare the libraries. After tagmentation, the cDNA was linked with two different index adapters from the Nextera XT Index Kit (Illumina) during the amplification by PCR. Each sample received a unique combination of two specific index/barcodes during the library generation. Finally, the libraries were purified using Agencourt AMPure XP beads that provide a size selection to remove short library fragments. Again, 1 μL of the libraries was used for validation using the Agilent 2100 Bioanalyzer on a High Sensitivity DNA chip. After preparation of the libraries, sequencing was performed by the Next Generation Sequencing Center (University of Pennsylvania, https://ngsc.med.upenn.edu/) on HiSeq 2500 machine.

**mRNA-seq data processing**. Following initial processing and quality control, the sequencing data was aligned to the mouse genome mm10 using Subread/featureCounts (v1.5.3; https://sourceforge.net/projects/subread/files/subread-1.5.3/) with an average read depth of 21,554,242 successfully aligned reads. Statistical analysis was carried out using Rsubread (v1.34.2; https://bioconductor.org/packages/release/bioc/html/Rsubread.html), DEseq2 (1.24.0; https://bioconductor.org/packages/release/bioc/html/DESeq2.html), and edgeR (v3.26.5; https://bioconductor.org/packages/release/bioc/html/edgeR.html), all in the R environment (v3.6.0; https://www.r-project.org/). Functional annotation clustering was performed using the Database for Annotation, Visualization and Integrated Discovery (DAVID v6.8, https://david.ncifcrf.gov). All GO terms were considered significant if *p* < 0.05 following corrections for multiple comparisons using the Benjamini–Hochberg method to control the FDR. *t*-Distributed stochastic neighbor embedding (tSNE) plots were generated to visualize patterns of similarity

between groups. To generate the tSNE, normalized data from all genes in the analysis were submitted and the perplexity set as high as possible given the total number of samples in the analysis with the theta set to 0.5.

**Single-cell isolation and library creation**. Separate biological replicates from AβE3 or AβE4 infused WT and Trem2$^{ko}$ mice were used for scRNA-seq and infused as described above. Twenty-four hours after the injections mice were perfused with PBS. Microglia from non-injected WT mice were used as negative control. After the brain extraction, cortical tissue around the injection site were excised using adult mouse brain matrix at 1 mm before and after the injection site. The brain tissue was then dissociated according to the manufacturer's instructions (Miltenyi Biotec) as described above. CD11b-purified cells were resuspended in 0.04% BSA/PBS solution and passed through a 40-μm nylon and a 20-μm nylon cell strainer to obtain single cell suspension. The cell viability and number were assessed using a Countess II FL Automated Cell Counter (ThermoFisher) with a Live/Dead viability/Cytotoxicity kit (Invitrogen). The single cell suspension with >90% viability was immediately loaded onto the Chromium controller (10x Genomics) in order to capture ~5000 cells per sample. Libraries were generated with Chromium single cell 3′ chip kit v3 for scRNA-seq (10x Genomics) according to the manufacturer's instructions and then the quality assessed using Agilent Bioanalyzer 2100. Sequencing was performed using NovaSeq (Illumina) by MedGenome Inc. with a custom sequencing setting (28 bp for Read1 and 91 bp for Read2) to obtain an average sequencing depth of 45 K reads/cell.

**scRNA-seq data processing**. After libraries were sequenced and quality control was performed, samples were aligned to the mm10 mouse reference genome using the Cell Ranger 3.0.1 pipeline (https://support.10xgenomics.com/single-cell-gene-expression/software/pipelines/latest/what-is-cell-ranger). Each sample was then aggregated using the cellranger aggr function to produce a raw UMI count matrix containing the number of reads for genes in each cell per sample. The expression matrix was then loaded into R for further analysis and visualization using Seurat v3.0.2[53]. Genes expressed in fewer than 200 cells were excluded from further analysis. The number of genes expressed in each cell (nGene), the number of reads in each cell (nCount), and the percentage of reads mapped to mitochondrial genes (percent.mito) were obtained for each cell. Cells were then filtered to reduce the potential of including doublet and low-quality cells using the following criteria: 200 < nGene < 8500; 500 < nCount < 90,000; and percent.mito <25%. Feature counts were normalized using LogNormalize method with a scale factor of 10,000; and the effects of nGene, nCount, and percent.mito were regressed out using the ScaleData method. A shared nearest neighbor (SNN) graph was constructed using FindNeighbors function with default parameters. Using the Louvain algorithm implemented in FindClusters function with a clustering resolution of 0.4 and the first 60 PCs, we identified 20 clusters. To determine the cell types, present in each cluster, we examined the expression levels of cell type specific markers across each cluster and identified clusters containing unique populations of microglia, neurons, astrocytes, vascular cells, ependymal cells, and myeloid cells. We used FindAllMarkers function to identify genes that act as markers for each cluster, using the Wilcoxon rank-sum test. A gene was considered the marker of a cluster if it had a Bonferroni-adjusted *p* value <0.01 and an average log fold change >0.1. The data were then filtered to contain clusters containing only microglia and cells with a normalized expression <1 for *Aldoc*, *F13a1*, *Mylk*, *Meg3*, *Gfap*, *Thy1*, *Slc17a7*, *Olfm1*, *Aqp4*, *Dlx1*, *Olig2*, *Foxj1*, *Nkg7*, *Vtn*, *Flt1*, *Acta2*, and *S100a9* were removed, resulting in the retention of 39,898 microglial cells. To control for the potential of batch effect we regressed out the percentage of UMI counts in each cell belonging to several sex-linked genes: *Ddx3y*, *Eif2s3y*, *Xist*, *Kdm5d*, and *Uty*. Using a clustering resolution of 0.15 and the first 50 PCs, we identified six separate clusters. The data were further filtered to contain only cells from injected samples, resulting in the retention of 36,244 microglial cells. After reclustering with a resolution of 0.1 and the first 50 PCs, we identified five clusters. To perform differential expression analysis, we used Seurat's FindMarkers function and performed Wilcoxon rank-sum tests. A gene was considered differentially expressed if it had a Bonferroni-corrected *p* value <0.05 and a natural log fold change (logFC) >0.1. Seurat data objects were reformatted for pseudotime analysis using the SingleCellExperiment package and trajectory analysis was performed using Slingshot[54] establishing cluster 1a as the starting cluster and clusters 2, 3, and 4 as terminal clusters.

**Fluorescence in situ hybridization (FISH)**. In a separate infused cohort, mice were perfused, tissue fixed, and sectioned as documented above. RNAscope experiments were performed using the Multiplex Fluorescent Reagent kit v2 (Advanced Cell Diagnostics) following the manufacturer's recommendations with minor adjustments. Six freshly sectioned tissues per animal were mounted onto superfrost plus slides within a 0.75″ x 0.75″ square and baked at 60 °C for 60 min. Slides were dehydrated using a series of ethanol dilution steps, then submerged in target retrieval reagent at 100 °C for 10 min. Protease digestion was performed at 40 °C for 30 min using Protease III and probe hybridization was carried out at 40 °C for 2 h. We used probe sets available from ACD for *Adgre1* and *Tmem119*. Following the amplification steps, the sections were counterstained with DAPI and coverslipped. Imaging was carried out using a Nikon Eclipse 90i microscope at 20X magnification with tiled imaging of the entire infusion site and analyzed using NIS

Elements software (Nikon Instruments Inc.). Four 200 μm x 200 μm ROIs were drawn and a threshold established for each probe to determine the area of puncta coverage. The four ROI were averaged together to create one data point per infusion site representing the average area of puncta coverage within 200 μm of the edge of the infusion site.

**Two-photon imaging and statistical analysis.** We used Cx3cr1[GFP] male mice (3 month) to determine the microglial response of AβE3 or AβE4 utilizing two-photon imaging[55]. Briefly, the animals were anesthetized I.P. with a cocktail of ketamine/xylazine (75/10 mg/kg) and placed in a stereotaxic frame. Body temperature was maintained at 37.6 °C. The skin above the skull was removed and a well of 2 cm diameter was built over both hemispheres with a light-curable cement (Composite Flowable) to hold a saline immersion for the microscope objective. The craniotomy was conducted with a high-speed dental drill over the parietal cortex bilaterally. The craniotomy site was regularly flushed with saline to wash out bone fragments and prevent thermal damage. The exposed cortex was injected with Hi-Lyte™ Fluor 555-labeled Aβ (Anaspec) combined with native E3 or E4 particles preincubated for 24 h. The injection was performed at an approximate depth of 300 μm below the surface of the cortex with a borosilicate glass capillary tube (WPI) pulled with a two-stage vertical pipette puller (Narishige) to a tip diameter of ~10 μm and coupled with flexible tubing to a picospritzer microinjection dispense system (Parker). Mice were randomly assigned to either the AβE3 or AβE4 group. After the injection, the mice were moved under the two-photon microscope within 5 min and anesthesia was maintained with 40 mg/ml/h ketamine.

In vivo imaging was conducted with an Ultima IV two-photon laser scanning microscope (Bruker) with an InSight DS Laser system (Spectra-Physics) tuned at a wavelength of 920 nm. Two channels were collected red (595/50) and green (525/50 nm) to visualize the 555-labeled Aβ and microglia, respectively. Images were collected with 16x Nikon objective lens on Prairie View software using a time series of 11 slices with 1024 × 1024 matrix size, dwell time 4.4 μs and 3 μm step size. The effective in-plane resolution was 0.4 μm per pixel. The Z-stack images were collected in Galvo acquisition mode such that each Z-stack was collected every minute for a total duration of 80 min.

For each animal, Z-stack images were analyzed to quantify the microglial approach and microglial injection site coverage over time. Initially, the data were loaded in ImageJ (National Institutes of Health) and a rigid-body co-registration with the StackReg plugin was used to account for global temporal drifts of the field of view. The Z-stacks were transformed to sum-based XY projections by linear interpolation using the 3D Project function, allowing for time series analysis of specific ROI. We generated the panels illustrating cell displacement from those datasets by assigning red to the beginning of the time series ($t = 0$) and green to subsequent timepoints (10 or 80 min). Then, the two color-coded images were merged in ImageJ to reveal spatial details, with red showing the original location, green for the new cellular location after a particular segment of time has elapsed (0–10 min or 0–80 min), and yellow at the regions where the two channels overlap. The 555-labeled Aβ infusion site was shown in white to display the increasing coverage as cell processes approach. The time series were also saved as movies presented in the supplementary material.

The datasets were then converted to mat structures and all subsequent analyses were performed in MATLAB (R2019a, Math Works), using standard image processing and statistical functions. Briefly, the red channel showing the infusion site was thresholded using the Otsu's method with the multithresh Matlab function. The resulting images were binarized and small features outside of the infusion site were removed using the bwareaopen function. The mask was then dilated to smooth the borders and account for the lower contrast of the diffusion gradient around the borders. The percent coverage by the microglial processes over time was then computed as the increase of the green channel intensity above 1.5 times the standard deviation baseline of the green channel within the binary mask.

The mean distance of the cell processes to the infusion site over time was computed by eroding the binary mask described above and computing the Euclidean distance transformation of the full binary image using the bwdist function. Then, each pixel in the green channel over 1.5 standard deviations of the intensity was mapped with that transformation and assigned a number that is the distance between that pixel and the edge of the infusion site forming progressive concentric level sets with increasing proximity to the infusion site. The change of this Euclidean distance over time was converted from pixels to microns using the effective in-plane resolution of the imaging data and plotted as a time series.

To verify Aβ biodistribution, Z-stack images had two color channels—red channel (595/50 nm) to capture the Aβ + APOE infusion and green channel (525/50 nm) to establish the initial positions of microglia within the infusion volume. The in-plane matrix size was 1024 × 1024 pixels with an effective resolution of 0.8 μ/pixel. We used a step size of 5 μ along the z direction across the cortical thickness, delivering an effective tissue volume of 3.251 μ³ per pixel. Each Z-stack was collected across the full depth of the infusion site with an average stack size of 30 slices and image analysis was performed in MATLAB. In order to quantify the three-dimensional distribution of the infusion, we converted the 16-bit red channel to binary matrix based on global foreground mean threshold (Otsu's method), eroded all small noise regions not connected to the infusion site and rendered the surface to verify consistent shape based on diffusion from the injection tip. We used the volume-per-pixel value (3.251 μ³) to calculate the total infusion volume

based on the number of non-zero matrix elements within the three-dimensional binary mask.

**Primary microglia cell culture.** Primary microglial culture and Aβ uptake assay were performed using WT, E3, E4, Trem2[ko], E3/Trem2[ko], and E4/Trem2[ko] pups (1–3 days old)[44,45]. The cortices and hippocampi were mechanically dissociated using a sterile Pasteur pipette. The dissociated cells were plated in 75 cm² flasks with DMEM/F12 medium (Thermo Fisher) containing 10% FBS at DIV0. Twenty-four hours after plating (DIV1), the media was replaced to remove debris. Microglia were collected by tapping at DIV14 and plated on 0.01% poly-L-lysine (Sigma) coated 12 mm circular coverslips in 24-well plates with the density of 60,000 cells/well. Twenty-four hours after plating, cells in the treatment groups were treated with 1 μM Hi-Lyte™ Fluor 488-labeled Aβ or Hi-Lyte™ Fluor 555-labeled Aβ (Anaspec) at 37 °C for 1 h. Following the treatment, cells were washed three times with PBS, fixed in 4% PFA, and permeabilized with 0.2% Triton X-100 at room temperature. Microglia were then labeled with Anti-IBA1 antibody (1:500) at 4 °C for 18 h and followed by 2 h incubation with horse anti-rabbit 594 secondary antibody (1:250) for 488-labeled Aβ or donkey anti-rabbit 488 secondary antibody (1:250) for 555-labeled Aβ. Microglia were stained with DAPI to visualize nuclei.

Fluorescent images of in vitro Aβ-treated microglia were taken on a Nikon Eclipse 90i microscope (20X magnification) and analyzed in NIS elements (Nikon Instruments Inc.). Exposure levels for each channel were consistent across all genotypes and samples. Images were thresholded to identify microglia, nuclei, and Aβ. To assess the Aβ uptake in microglia, the percentage of microglia containing Aβ was determined by dividing IBA1 + microglia colocalized with Aβ signal by the total of IBA1 + microglia and averaged across all images in each genotype.

**Quantification and statistical analysis.** Sample sizes ($n$) indicated in the figure legend 1 indicate number of separate pools of astrocyte conditioned media. Sample sizes ($n$) indicated in the figure legend 2–7 correspond to the number of biological replicates analyzed. All researchers were blinded to experimental groups during the analysis. All results are reported as means ± SEM.

Unless otherwise indicated, all statistical analyses were performed in GraphPad Prism (v 8.3.1), or in R (v 3.6.0) and significance was determined as $p < 0.05$. Number of experiments and statistical information are stated in the corresponding figure legends. In figures, asterisks denote statistical significance marked by $*p < 0.05$; $**p < 0.01$; $***p < 0.001$.

**Reporting summary.** Further information on research design is available in the Nature Research Reporting Summary linked to this article.

## Data availability
The bulk RNA-seq and single cell RNA-seq data in this study are available in the NCBI Gene Expression Omnibus (GEO) database under the accession number: GSE158962 (https://www.ncbi.nlm.nih.gov/geo/query/acc.cgi?acc=GSE158962). All of the sequencing datasets used in this study were processed using publicly available open-source software. The sequencing data was aligned to the mouse genome mm10 (https://bioconductor.org/packages/release/data/annotation/html/org.Mm.eg.db.html). Functional annotation clustering was performed using the Database for Annotation, Visualization and Integrated Discovery (DAVID v6.8, https://david.ncifcrf.gov). Source data are provided with this paper. All data supporting the findings of this study are provided within the paper and its supplementary information. Additional information is available upon reasonable request to the authors. Source data are provided with this paper.

## Code availability
Bulk RNA-seq and single cell RNA-seq data were processed using publicly available open-source software and standard pipelines were used, as described in the Methods. The Rsubread (v1.34.2), DEseq2 (v1.24.0), and edgeR (v3.26.5) were used for bulk RNA-seq data analysis. The Seurat (v3.0.2) and Slingshot (v1.8.0) were used for single cell RNA-seq data analysis. All relevant code is available from the authors upon reasonable request.

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

## Acknowledgements

This research was supported by the National institute of Health, grant number AG056371, AG057565, AG066198 (I.L. and R.K), NS116450 (B.E.I.), and NS094396 (T. D.Y.K.) and the Alzheimer's Association, 2018-AARG-590509 (N.F.F.) and AARF-16-443213 (K.N.N.). Human brain samples were provided by ADRC Pittsburgh (P30AG066468) and Dr. Peter Nelson, University of Kentucky ADRC (P30 AG028383). We thank all members of Lefterov/Koldamova lab for their excellent technical assistance.

## Author contributions

N.F.F., I.L., and R.K. conceived and designed the research and wrote the manuscript. All authors edited the drafts. N.F.F. and B.E.P. set up and performed mouse experiments, IHC, and Aβ aggregation experiments. C.M.W. and K.N.N. isolated APOE lipoproteins from ACM. K.N.N. performed, analyzed, and wrote the single cell RNA sequencing experiments, and in vitro experiments. C.M.W. and F.L. performed flow cytometry and bulk sequencing experiments and analysis. C.M.W. performed FISH experiments. R.J.B. participated in the analysis of the single cell RNA sequencing data. B.E.I. performed, analyzed, and wrote the two-photon imaging experiments. V.E.K., Y.Y.T., and X.H. performed lipidomics experiments and analysis. T.D.Y.K. provided essential resources.

## Competing interests

The authors declare no competing interests.
