## [Peer Review File · Nature Communications]

Reviewers' Comments:

Reviewer #2:

Remarks to the Author:

Review of Fitz et al. Nature Communications manuscript

PHOSPHOLIPID COMPOSITION OF APOE LIPOPROTEINS AFFECTS 1 MICROGLIA ACTIVATION IN AN ISOFORM-SPECIFIC MANNER

In this manuscript, Fitz and colleagues examine the lipid signature of purified apolipoproteins E3 and E4 secreted from primary mouse astrocytes using lipidomic techniques and found some similarities with that of post-mortem tissue from individuals with the ApoE3 or ApoE4 variants, namely a deficiency in multiple families of phospholipids. The authors then inject preparations of amyloid-beta oligomers and ApoE3 vs ApoE4 particles into the mouse CNS and find that Abeta-ApoE3 mixtures are less deleterious than Abeta-ApoE4 mixtures in cognitive tests. Consistent with an involvement of microglia in these responses, they show that ApoE3 lipoproteins are more efficacious at inducing a chemotactic response of microglia toward Abeta compared to ApoE4 lipoproteins. They also examine the microglial transcriptome in response to A β using bulk and single cell RNA-seq analysis of wild-type and Trem2 KO brains and find that ApoE3 lipoproteins upregulate DAM genes more efficiently than ApoE4 lipoproteins, but not in the Trem2 KO microglia. This is consistent with previous studies from multiple groups showing that TREM2 ablation locks microglia in a more homeostatic state.

While it makes a lot of sense to study the impact of ApoE4 on lipid metabolism and its relationship with TREM2 LOF, this ambitious study embarked on a path that makes interpretations very difficult, in part through the use of an experimental paradigm that is largely artificial, namely intracerebral injections of Abeta oligomers bound to cell derived apolipoproteins. The main concerns are as follows:

1. While the Abeta/lipoprotein complexes have interesting pharmacodynamic effects in vivo, there is no attempt to determine whether these effects relate to the levels of apolipoproteins or Abeta oligomers injected. The authors should provide evidence showing that the levels and biodistribution of Abeta/lipoprotein complexes are comparable for ApoE3 vs ApoE4 and examine Abeta levels in situ post-injection.
2. The authors compare the lipid composition of ApoE3 and ApoE4 lipoprotein particles with post-mortem brain tissue from human subjects with E3 or E4 variants. The better comparison would be to examine the lipid composition of CSF from human E3 and E4 carriers and see whether it matches that of purified E3 vs. E4 apolipoproteins.
3. Similarly, it is unclear why the authors do not leverage the ApoE3 and ApoE4 mice to determine if the brain phospholipid composition of ApoE4 mice shows the same deficiency as that found in purified ApoE4 lipoproteins.
4. From a mechanistic point of view, the authors do not demonstrate a link between the protective effect of ApoE3 lipoproteins and their higher phospholipid content compared to ApoE4 lipoproteins. If they purify ApoE lipoproteins from HEK293 cells, which are known to be much less lipidated than astrocyte-derived lipoproteins, do they lose the isoform specific differences in the pharmacodynamic effect? Alternatively, if they inject ApoE3 apolipoprotein particles derived from the ABCA1 mutant astrocytes, do they phenocopy the effect of the ApoE4 apolipoproteins?
5. Are the reduced levels of phospholipids found on ApoE4 vs. ApoE3 lipoproteins the molecular basis for lack of response of microglia? If so, is it because these phospholipids bind to TREM2? This could be modeled in vitro with primary microglia from WT and Trem2 KO mice treated with ApoE3 vs. ApoE4 lipoproteins.
6. The authors purify lipoproteins from ApoE3 and ApoE4 expressing astrocytes, but these are very crude preparations and it is unclear whether other apolipoproteins released by these cells (or other factors) contribute to the isoform-specific phenotypes. Can the authors use ApoE KO astrocytes as a source of other lipoproteins as a control?
7. Human astrocytes expressing ApoE4 show cholesterol phenotypes compared to those expressing ApoE3 (PMID: 29953873). Additionally, mouse ApoE KO astrocytes and microglia show an upregulation of cholesterol metabolites in vivo (for instance, esters of cholesterol) (PMID: 31902528). This strongly suggests that ApoE controls cholesterol metabolism. What are the levels

of cholesterol and cholesterol esters in the purified lipoproteins from ApoE3 and ApoE4 astrocytes?

Reviewer #3:

Remarks to the Author:

The interesting paper by Fitz et al uses amyloid-ApoE isoform co-injections coupled with in vivo 2 photon imaging, behavioural testing and RNA-seq analyses to examine the effect of ApoE isoform on functional and molecular properties of microglia and the interactions between TREM2 and ApoE isoform.

The authors showed widespread reductions of ~50 species of phospholipids in ApoE4/4 lipoproteins in astrocyte-conditioned media and ApoE4/4 patient brains, and that injection of lipoproteins to mice prevented amyloid-induced impairments in contextual fear conditioning and novel object recognition.

The experiments appear to be well performed, properly controlled and adequately powered for behavioural analyses. This work is novel, exciting and adds to the body of literature examining why ApoE4 poses increased risk for Alzheimer's. The results on the interaction between ApoE isotype and Trem2 in microglia are particularly interesting. The authors show that microglia exposed to E4 lipoproteins cannot activate the phagocytic program as early or well as microglia exposed to E3, and Trem2 regulates the early transition of microglia.

The presentation of figures could be more clear and consistent, and more details in the figure legends would aid the reader's understanding.

Comments:

1. Other than DAM, are the E3 or E4 induced microglia signatures similar to any other previously identified signatures in mouse and/or human AD (ie MgND, Neurodegeneration, LAM, IFN signatures etc)? In particular, please discuss the E4-induced signature in more detail, as this may be relevant to how ApoE4 increases AD risk, and compare to the signature shown in human ApoE4 iPS-derived microglia (Lin et al 2018 Neuron) and in mouse ApoE4 microglia (Zhao et al 2020 Neuron). Similarly, are the early/late signatures and single/dual signatures unique, or similar to previously described microglia?
2. The RNA-seq data in Fig 2 do not support the conclusion that "A β E3 induce stronger microglial response when compared to A β E4" – there are a similar number of DEGs upregulated/downregulated in both groups and a number of processes perturbed.
3. The FACS plots Fig3c do not visually look to be different between E3 and E4, except that E4 look to have slightly higher fluorescence intensity for Abeta-555 (same seems to be true in Fig S3b-d). Could these be represented as a contour plot to make differences more visible to the reader and could the authors please calculate the MFI of Abeta-555 in dual positive cells? Is it possible that more E3 cells take up Abeta but each individual E4 cell takes up more Abeta? Or that phagocytic flux is reduced in E4 and thus more Abeta accumulates? Could the authors please discuss their findings in the context of Muth et al 2019 Front Cell Neurosci, who found that ApoE4 promotes microglial cell motility and upregulates phagocytosis of apoptotic cells, although also found reduced capacity for A β uptake. The authors work supports that of Konttinen et al 2019 Stem Cell reports, who also found reduced migration and phagocytosis, but enhanced cytokine production in APOE4 iPS derived microglia like cells, and could be discussed.
4. The authors suggest that a rapid response by E3 microglia to isolate A β may be a protective mechanism diminishing the spread of A β . This should be relatively straightforward to test using their paradigm and assessing A β spread at a later time point. Similarly, later timepoints could be used to assess the efficiency of plaque coverage by microglia.
5. Can the authors attempt to functionally define each of the 5 microglial clusters? ie Cluster 1a is almost entirely composed of cells from mice infused with E3 lipoproteins- thus even "homeostatic" microglia in an E4 context are not quite "homeostatic".

6. Could the authors please comment on E3 function: is E3 preventing oligomer formation and/or inducing stronger microglia responses or both and which is likely to be more important in the brain?

Minor comments:

1. Line 48- species not spices

2. Please clarify whether Fig 1a is from previous or current paper – should be referenced not included if the former?

3. Fig 1e – the text is too small to read, it's not clear what is being shown, what the axes represent, what the grey bars represent and why there are 2 bars for most phospholipids, but 3 bars for SM. Are correlations in all samples or in ApoE3/3 or 4/4 samples only? More detail in the figure legend would help clarify these points

4. Fig 1f – please include a legend to show what the axes represent

5. Fig 1f-h to clarify, are these analyses performed on APOE3/4 lipoproteins isolated from astrocyte-conditioned media? If so, please state more explicitly in methods/results or figure legend

6. Fig S1b-j please label the x axes

7. Fig S2a – please provide inset images for all panels as the structure of oligomers is difficult to see

8. Fig 2 – please ensure that all text sizes are consistent

9. Fig 2 b,d,I,j, please include all datapoints as dot plots to make the n and data distribution clearer for the reader

10. Fig 2 ij it's not clear what is being measured here as the heatmap images do not look similar to the Iba1 staining images below. Please clarify what the upper images and % coverage refers to – does this correspond to the individual microglia below or the region?

11. Fig 2lm the data appear to be from 7 mice per group, was one mouse excluded and why?

12. Fig 4a –microglia misspelt microglai

13. Fig 4 –keeping the colour scheme for each of the 8 populations consistent would really help visual interpretation of this data e.g. fig 4a v fig4b

14. Fig 4e – please show the error bars for CPM values for the replicates of gene expression

15. Fig 5f – are differences significant?

16. Line 350: "we" should be "as"

Reviewer #4:

Remarks to the Author:

In this paper, Fitz et al demonstrate a role for APOE genotype in determining cognitive outcomes and microglial phenotypes following A β infusion into the brain. Using in vivo 2-photon imaging, the authors provide evidence to show that APOE genotype affects microglia mobilization towards A β oligomers. The authors also demonstrate that APOE genotype affects microglial transcriptional programs, then investigate the potential interaction between APOE genotype and TREM2. While these data are certainly valuable, the manuscript contains several major weaknesses. The TREM2 data presented in this manuscript lack novelty, and several conclusions made by the authors

require further support. In addition, the different areas of the paper are not well connected to each other. Therefore, as discussed below, I think this paper in the current form has not provided sufficiently significant conceptual advances to warrant its publication in this journal.

Major Comments

1. Different parts of the manuscript are somewhat disconnected from each other. While compelling, the lipidomic data in Figure 1 are not directly relevant to the remainder of the paper. In addition, while Figures 2-4 discuss a potential relationship between APOE isoform and microglial function, Figures 5-7 shift towards discussing the effect of TREM2 on DAM gene expression. As presented, these stories are not well integrated to make coherent conclusions.

2. Figures 5-6 focus on the role of TREM2 on DAM gene expression. However, it is well accepted in the field that TREM2 is necessary for homeostatic microglia to transition to the DAM state. These data lack novelty and distract from the other more innovative experiments in this manuscript.

3. The authors have not included a control condition (non-infused or mock-infused) in their single-cell sequencing experiments. This is particularly problematic since the authors conclude that Clusters 2, 3, and 4 appear in response to A β infusion. Without a control, it is not possible to identify which of these populations actually result from A β infusion. Previous scRNA-seq studies have shown that DAM (Cluster 2) and interferon-related (Cluster4) populations are also present in the wildtype brain without treatment. In addition, Cluster 3 is characterized by expression of immediate early genes, which can be induced by experimental artifacts resulting from certain methods of cell isolation (Haimon et al. Nat Immunol. 2018)— based on the presented data alone, it is unclear how different microglial populations change in response to A β -APOE.

4. The authors correlate changes in microglia morphology and gene expression with the behavioral outcomes observed in Figure 2, suggesting a relationship between microglia and APOE3-mediated cognitive improvement. However, the involvement of microglia in this behavior is unclear. The authors should provide more causal data to substantiate this suggestion.

5. Disease-associated microglia are thought to constitute a restricted subset of microglia that upregulate a set of signature genes. However, the authors compare DAM gene expression for microglia from all three Clusters 2, 3, and 4 (Figure 6a-d). It is unclear how such comparisons can be interpreted based on the conventional understanding of DAM. What is the significance of this gene expression pattern in clusters that are dissimilar to DAM?

6. Figure 7 does not effectively illustrate the interaction between TREM2 and APOE genotype on microglia gene expression. It is difficult to arrive at the conclusions reached in the text based on this figure alone. It would help to first present the interaction between TREM2 and each isoform separately, before comparing these interactions to each other as shown in Figure 7.

Minor Comments

1. To strengthen the impact of the single-cell sequencing data, the authors should clarify how many replicates were performed for this experiment. The legend for Figure 5 states that 2 mice were used per group, but was the whole sequencing experiment repeated? This repetition is important, since the authors conclude that cluster sizes differ among wild-type and TREM2 knockout microglia.

2. The header of the first results section could benefit from being reworded. It not immediately clear that the two groups being compared are the human lipids and the native lipoproteins.

3. Figure 1E is difficult to interpret. To aid understanding, the authors might consider including some additional guidance in the legend or some labels on the figure.

4. It is unclear where the genes listed on Figure 3 are coming from at this point in the paper, since the authors discuss transcriptional data in Figure 4.

5. The FACS data in Figure 3C do not look representative of the bar plot quantifying these data.

The quantification shows a ~1.5 to 2-fold increase in dual-positive cells in the ApoE3 condition, but visually this is difficult to appreciate.

6. In the legend for Figure 3D, the authors state that early time points are depicted in green while late timepoints are depicted in red. However, the opposite is stated in the text (line 185).

7. While meant to be a comparison between A β E3 and A β E4, the data presented in Supplemental Figures 3E and 3F are identical.

Major changes of the manuscript

1. We included additional lipidomic data from brains of APOE4 carriers (added to Fig. 1a).
2. We included phospholipid measurement on cerebrospinal fluid (CSF) collected from APP mice expressing human APOE3 or E4 (referred to as APP/E3 and APP/E4) on Suppl. Fig. 1.
3. We performed measurement of cholesterol esters now added to Suppl. Fig. 1 in the revision.
4. We included new data on microglia barrier surrounding amyloid plaques in APP/E3 and APP/E4 mice (Suppl. Fig. 3).
5. We did perform 2 additional sets of *in vitro* experiments. In the first set, we used microglia isolated from WT pups treated with native APOE3, APOE4 and APOE3^{het} and A β alone as negative controls (Suppl. Fig. 4). In the second set we used microglia isolated from WT or TREM2^{ko} pups treated with native APOE3 or APOE4 (added to Fig. 7).
6. To assess the injected volume and biodistribution of the injected A β we did additional analysis of two photon imaging data in MATLAB (added to Suppl. Fig. 5).
7. We performed single cell RNA-seq on WT non-injected mouse as a negative control to the injections and included this control in the analysis of the scRNA-seq (added to Fig. 5 and Suppl. Fig. 7 and 8).
8. We reanalyzed scRNA-seq data to include comparisons of WT vs Trem2^{ko} microglia for each treatment (WT microglia injected with A β E3 to Trem2^{ko} microglia injected A β E3 and the same for A β E4 injection).
9. We added additional comments as a response to the reviewers' critiques. We corrected mistakes according to the reviewers' minor critiques.
10. In the text the results related to the new figures and other edits are indicated with red font color highlighting.
11. High resolution copies of figures and tables can be found and downloaded through figshare at the following locations:
 - a. Main text figures: <https://figshare.com/s/e8c73891e6a1e2c370dc>
 - b. Supplemental figures: <https://figshare.com/s/9f2d7daf9f5e697b265a>
 - c. Supplemental tables: <https://figshare.com/s/9382e9185970c7f806b8>

Reviewer 2

In this manuscript, Fitz and colleagues examine the lipid signature of purified apolipoproteins E3 and E4 secreted from primary mouse astrocytes using lipidomic techniques and found some similarities with that of post-mortem tissue from individuals with the ApoE3 or ApoE4 variants, namely a deficiency in multiple families of phospholipids. The authors then inject preparations of amyloid-beta oligomers and ApoE3 vs ApoE4 particles into the mouse CNS and find that A β -ApoE3 mixtures are less deleterious than A β -ApoE4 mixtures in cognitive tests. Consistent with an involvement of microglia in these responses, they show that ApoE3 lipoproteins are more efficacious at inducing a chemotactic response of microglia toward A β compared to ApoE4 lipoproteins. They also examine the microglial transcriptome in response to A β using bulk and single cell RNA-seq analysis of wild-type and Trem2 KO brains and find that ApoE3 lipoproteins upregulate DAM genes more efficiently than ApoE4 lipoproteins, but not in the Trem2 KO microglia. This is consistent with previous studies from multiple groups showing that TREM2 ablation locks microglia in a more homeostatic state. While it makes a lot of sense to study the impact of ApoE4 on lipid metabolism and its relationship with TREM2 LOF, this ambitious study embarked on a path that makes interpretations very difficult, in part through the use of

an experimental paradigm that is largely artificial, namely intracerebral injections of Abeta oligomers bound to cell derived apolipoproteins.

The main concerns are as follows:

1. While the Abeta/lipoprotein complexes have interesting pharmacodynamic effects *in vivo*, there is no attempt to determine whether these effects relate to the levels of apolipoproteins or Abeta oligomers injected. The authors should provide evidence showing that the levels and biodistribution of Abeta/lipoprotein complexes are comparable for ApoE3 vs ApoE4 and examine Abeta levels *in situ* post-injection.

We injected the same infusion volume for all animals. To assess the injected volume and assess the biodistribution, post-injection we acquired *in vivo* three-dimensional z-stacks. Z-stacks taken to verify the biodistribution had two color channels - red channel (595/50 nm) to capture the spread of the A β E3/ A β E4 or infusion and green channel (525/50 nm) to establish the initial position of the microglia within the injected volume. We segmented, rendered and computed the three-dimensional volumes of each infusion site in MATLAB. The shape and injection volume of A β E3 and A β E4 groups did not differ significantly (t test, alpha level 0.05, p=0.6). This analysis was added to supplemental Fig. 5d, to the method section and is referred to in the results.

2. The authors compare the lipid composition of ApoE3 and ApoE4 lipoprotein particles with post-mortem brain tissue from human subjects with E3 or E4 variants. The better comparison would be to examine the lipid composition of CSF from human E3 and E4 carriers and see whether it matches that of purified E3 vs. E4 apolipoproteins.

3. Similarly, it is unclear why the authors do not leverage the ApoE3 and ApoE4 mice to determine if the brain phospholipid composition of ApoE4 mice shows the same deficiency as that found in purified ApoE4 lipoproteins.

As these questions are related we combined our answer.

We used pooled CSF samples from APP/E3 and APP/E4 mice to measure the total phospholipids level. The new data shown in the re-submission of Supplemental Fig.1k, demonstrates that phospholipid concentration in CSF is higher in APP/E3 mice when compared to APP/E4. As we note in the article, the native APOE-containing lipoproteins isolated from astrocyte conditioned media are reminiscent of the lipoproteins that circulate in the brain interstitial fluid (ISF). It should be noted that, whereas CSF is in communication with the ISF, these two fluids are not the exactly the same. In terms of the lipoproteins, CSF lipoproteins are significantly different from ISF lipoproteins in that they are spherical and contain additional apolipoproteins. As reviewed by Cheryl Wellington group^{1, 2}, CSF lipoproteins are spherical particles because they contain a cholesterol ester core. In contrast, APOE-containing lipoproteins in ISF and ACM are discoidal because they contain very little to no cholesterol esters. Furthermore, CSF lipoproteins contain, in addition to APOE and CLU (clusterin), other apolipoproteins such as APOA-I, APOA-II and APOA-IV that are not expressed in the brain¹. One possible reason for these differences is that CSF-blood barrier can allow the influx from blood of some small HDL particles via transcytosis¹.

In the discussion of the manuscript, we commented on the limitations of a comparison between lipoproteins isolated from ACM to brain parenchymal tissue: “*It should be noted that the human brain samples include lipids that are structural part of cellular membranes as well as lipoproteins in interstitial fluid*”. A comparison with mouse brain lipidomic could have the same limitations but lacking the advantage of being relevant to human disease. We believe that, regardless that the source of the phospholipids is not entirely the same in human brain parenchyma and ACM, it is still remarkable that there is a very high level of similarity in the direction of fold change between the phospholipids of human samples and the native lipoproteins that are APOE isoform-specific. To strengthen this conclusion, in the resubmission we added results from APOE4 carrier (APOE3/4) brain samples that were also a part of our published study³. As shown in the resubmitted manuscript (Fig. 1a), several

of the phospholipid classes (PE, PI, PS, SM and PA) demonstrate an apparent APOE3-allele dose-response effect in the order E3/3>E3/4>E4/4.

4. From a mechanistic point of view, the authors do not demonstrate a link between the protective effect of ApoE3 lipoproteins and their higher phospholipid content compared to ApoE4 lipoproteins. If they purify ApoE lipoproteins from HEK293 cells, which are known to be much less lipidated than astrocyte-derived lipoproteins, do they lose the isoform specific differences in the pharmacodynamic effect? Alternatively, if they inject ApoE3 apolipoprotein particles derived from the ABCA1 mutant astrocytes, do they phenocopy the effect of the ApoE4 apolipoproteins?

Figure 1 (Supplemental Figure 4 in the manuscript). Aβ uptake is differentially affected by native APOE lipoproteins *in vitro*. (supplemental to Figure 3)

555-labeled Aβ was pre-incubated with APOE native particles or PBS as indicated in the methods. Primary microglial cultures established from WT pups were treated for 1 h with Aβ+PBS (Aβ), AβE3 or AβE4 followed by IBA1 and DAPI immunostaining. Aβ pre-incubated with lipoproteins isolated from E3/Abca1^{het} astrocytes (AβE3^{het}) was used as negative control. Representative images from microglia incubated with Aβ only (a), AβE3^{het} (b), AβE3 (c) or AβE4 (d) with IBA1 in green, 555-labeled Aβ in red, and DAPI in blue. Scale bar=100 μm. (e) Bar plot showing the percent of microglia (IBA1) which co-localize to Aβ signal (555-label). Analysis by one-way ANOVA followed by Tukey's multiple comparison test. N=3 independent cultures per experimental group. Bars represent mean ± SEM. * p<0.05; ** p<0.01; *** p<0.001, NS, not significant.

We agree with the reviewer's comment and in the revision, we included experiments with lipoproteins derived from E3/Abca1^{het} astrocytes. We did not use HEK293 cells as they are not of brain origin and may express genes which are not expressed in the brain. As visible in the first submission, on Suppl. Fig. 1a, as negative control for lipidation we used lipoproteins derived from E3/Abca1^{het} astrocytes (referred to as E3^{het}) and the phospholipid content is E3>E4>E3^{het}. At the reviewer's suggestion we performed additional *in vitro* experiments utilizing WT microglia treated with Aβ alone, AβE3, AβE4 and used as negative control AβE3^{het}. These new data are presented in on Supplemental Fig. 4 and demonstrate that there is a dose-response effect on Aβ uptake by microglia in the following order AβE3> AβE4=AβE3^{het}>Aβ that is in correlation with the phospholipid level of the respective lipoproteins. This experiment also shows that the addition of even less lipidated APOE (such as AβE3^{het}) improves the uptake compared to Aβ alone.

It has been established that APOE isoforms' lipidation status is as follows: APOE2>APOE3>APOE4 (reviewed in⁴). Previous study from Gary Landreth and our groups^{5,6} demonstrated that APOE lipidation has a direct effect on Aβ degradation by microglia and is APOE isoform-dependent (reviewed in⁷). Furthermore, it has been shown that LXR/RXR ligands promote Aβ uptake and degradation by microglia by means of increasing APOE lipidation (^{5,6}, also reviewed in^{4,8}). In the current manuscript, we extend the prior research to examine the phospholipid content of APOE3 and APOE4 lipoproteins and demonstrate that there is a significant APOE isoform-dependent difference in the level of four phospholipid classes that correlates to the differential microglial response.

5. Are the reduced levels of phospholipids found on ApoE4 vs. ApoE3 lipoproteins the molecular basis for lack of response of microglia? If so, is it because these phospholipids bind to TREM2? This could be modeled *in vitro* with primary microglia from WT and Trem2 KO mice treated with ApoE3 vs. ApoE4 lipoproteins.

There were several reports that TREM2 binds to lipids, circulating lipoproteins and phospholipids. As the reviewer suggested, we performed additional experiments using primary microglia established from WT and Trem2^{ko} mice that were treated with AβE3 or AβE4. It should be noted that both WT and Trem2^{ko} microglial cells express mouse *ApoE* established from mice with the same genotype used for the *in vivo* experiments in the article (Fig.5-7). As shown on the Fig. 2 in this response and on Fig. 7e in the manuscript, in both WT and Trem2^{ko} cells there is a significant difference in the uptake of cells treated with AβE3 vs AβE4. In addition, there is a significant difference in Aβ uptake between WT and Trem2^{ko} cells only in AβE4 treated microglia. Finally, there is no difference between uptake of WT and Trem2^{ko} cells treated with AβE3 suggesting that APOE3 could compensate for the insufficiency of Trem2 microglia regarding the uptake.

Figure 2. Trem2 deficiency and APOE isoform affect microglia response to Aβ.

(a) 555-labeled Aβ was pre-incubated with APOE3 or E4 native particles. Primary microglial cultures were established from WT or Trem2^{ko} pups expressing mouse *ApoE*. Bar plot showing the average percentage of Aβ-containing microglia for each treatment. (b) Microglia from E3, E3/Trem2^{ko}, E4, and E4/Trem2^{ko} mice were treated with Aβ for 1 hour. Cells were labeled with IBA1 (red) and 488-labeled Aβ (green) from three independent cultures in triplicate for each genotype. Bar plot showing the average percentage of Aβ-containing microglia for each genotype. Analysis by one-way ANOVA followed by Tukey's multiple comparison test. Bars represent mean ± SEM. * p<0.05, ** p<0.01; *** p<0.001, N.S. no significance

This experiment is different from the experiment shown on Fig. 7e in the original submission: a) in the experiment shown in the original submission, we used microglia expressing human APOE isoforms and Trem2^{wt} (referred in the articles to as E3 and E4) or microglia from Trem2^{ko} mice again expressing APOE3 or E4 (E3/Trem2^{ko} or E4/Trem2^{ko}). b) we used Aβ alone without the addition of native E3 or E4. This experiment is now shown on Suppl. Fig. 10. We could speculate that the addition of native E3 or E4 lipoproteins serves as a ligand to TREM2 or other microglial receptors and facilitates Aβ uptake which is restored to normal with native E3 lipoproteins.

6. The authors purify lipoproteins from ApoE3 and ApoE4 expressing astrocytes, but these are very crude preparations and it is unclear whether other apolipoproteins released by these cells (or other factors) contribute to the isoform-specific phenotypes. Can the authors use ApoE KO astrocytes as a source of other lipoproteins as a control?

We agree with the reviewer that other apolipoproteins could affect Aβ uptake and microglia transcriptome. The only other apolipoprotein that is expressed in astrocytes and is included in the native APOE-containing particles is clusterin (CLU/APOJ) and it has been shown to bind TREM2⁹. However, the goal of our study was to compare two APOE isoforms, namely E3 and E4 in terms of the phospholipid content of their native lipoproteins and test how this affects Aβ uptake and microglia transcriptome and function. To achieve this goal, it was essential that E3 and E4 isoforms are present in order to compare the lipid level of their native particles and consequently the effect on microglia. As we showed in the answer to question #4 above, in order to vary the lipid component, we used E3 and E4 lipoproteins isolated from E3/Abca1^{het} or E4/Abca1^{het} which contain significantly less phospholipids than their Abca1^{wt} counterparts, but both APOE isoforms (APOE3 and E4) are present. CLU/APOJ is present in all of the native particles at the same level and otherwise we kept all other factors the same. As we mentioned above, for a negative control of Aβ uptake *in vitro* we used only E3/Abca1^{het} native particles.

7. Human astrocytes expressing ApoE4 show cholesterol phenotypes compared to those expressing ApoE3 (PMID: 29953873)¹⁰. Additionally, mouse ApoE KO astrocytes and microglia show an upregulation of cholesterol metabolites *in vivo* (for instance, esters of cholesterol) (PMID: 31902528)¹¹. This strongly suggests

that ApoE controls cholesterol metabolism. What are the levels of cholesterol and cholesterol esters in the purified lipoproteins from ApoE3 and ApoE4 astrocytes?

In addition to measuring phospholipids we also measured total cholesterol level of APOE-associated lipoproteins (presented on Suppl. Fig. 1) and found a small but significant difference in cholesterol content between E3 and E4 preparations. This assay measures total cholesterol, including free and esterified cholesterol. As we mentioned in the answer to question #2, APOE lipoproteins secreted by astrocytes contain very few cholesterol esters. Regardless, we measured cholesterol ester and it appears that there is only a small percentage (less than 10%) of cholesterol ester in the ACM samples and there is no difference between genotypes. This data are presented on Suppl. Fig. 1m.

Our study is significantly different from the two articles cited by the reviewer in that they examine intracellular cholesterol or cholesterol in the media which is not associated with APOE-containing lipoproteins. Our method to isolate native lipoproteins from astrocytes conditioned media uses de-lipidated serum for cell culturing. The method involves series of centrifugations (to remove dead cells and debris) followed by few size-exclusion steps and washing with PBS to retain relatively large particles (between 8-12 nm diameter) in which the cholesterol is incorporated in the APOE-containing lipoproteins. Thus, the non-lipoprotein associated cholesterol which may be found in the conditioned media as a media component or as a result of cell debris or other cell related process are eliminated.

In the first citation by Lin et al.¹⁰, the authors employ iPSCs cells to test APOE4 isoform effect on microglia. They conclude that “*APOE4 microglia-like cells exhibited altered morphologies, which correlated with reduced A β phagocytosis*” a conclusion reached by us in the current manuscript. Lin et al.¹⁰ measured intracellular cholesterol and cholesterol in the medium without specifying its source that makes it difficult to compare with our study.

The other article the reviewer is referring to, is a recent study by Nugent et al.¹¹ that uses a model of chronic demyelination and reports an increased accumulation of cholesterol esters in the brains of Trem2^{ko} mice. The authors conclude that this accumulation of cholesterol esters is primarily intracellular resulting from phagocytosis of cholesterol-rich myelin by microglia and storage in lipid droplets after conversion of cholesterol to CE by ACAT. The effect is abrogated by treatment with LXR agonist which increases the expression of Abca1 transporter. Most importantly, Nugent et al.¹¹ demonstrate that liposomes containing phospholipid classes, shown by us in this manuscript to be different in APOE3 and E4 lipoproteins such as SM, PI and PE, activate downstream Trem2 signaling. Unlike their study, however, we examined the effects of APOE isoform on the lipidome of human AD brain and ACM isolated lipoproteins and not in the brain of Trem2^{ko} mice.

Reviewer 3

The interesting paper by Fitz et al uses amyloid-Apoe isoform co-injections coupled with in vivo 2 photon imaging, behavioural testing and RNA-seq analyses to examine the effect of Apoe isoform on functional and molecular properties of microglia and the interactions between TREM2 and Apoe isoform.

The authors showed widespread reductions of ~50 species of phospholipids in Apoe4/4 lipoproteins in astrocyte-conditioned media and Apoe4/4 patient brains, and that injection of lipoproteins to mice prevented amyloid-induced impairments in contextual fear conditioning and novel object recognition.

The experiments appear to be well performed, properly controlled and adequately powered for behavioural analyses. This work is novel, exciting and adds to the body of literature examining why Apoe4 poses increased risk for Alzheimer's. The results on the interaction between Apoe isotype and Trem2 in microglia are particularly interesting. The authors show that microglia exposed to E4 lipoproteins cannot activate the phagocytic program as early or well as microglia exposed to E3, and Trem2 regulates the early transition of microglia.

The presentation of figures could be more clear and consistent, and more details in the figure legends would aid the reader's understanding.

Comments:

1. Other than DAM, are the E3 or E4 induced microglia signatures similar to any other previously identified signatures in mouse and/or human AD (ie MgND, Neurodegeneration, LAM, IFN signatures etc)? In particular, please discuss the E4-induced signature in more detail, as this may be relevant to how Apoe4 increases AD risk, and compare to the signature shown in human Apoe4 iPS-derived microglia (Lin et al 2018 Neuron) and in mouse Apoe4 microglia (Zhao 2020). Similarly, are the early/late signatures and single/dual signatures unique, or similar to previously described microglia?

In addition to DAM shown in Keren-Shaul et al. study¹², we compared our data to published studies such as Hammond et al.¹³ and Krasemann et al.¹⁴ that are most relevant to ours. We added these comparisons to the discussion.

- a. We found highly overlapped genes between dual+ microglia shown on Fig. 4 and injury responsive microglia (IRM) from Hammond et al.¹³, which contains lots of interferon signaling related genes. We found that the number of overlapped genes is higher in dual+ A β E3 at 4 hr than dual+ A β E4, suggesting E3 induces the change of the microglial phenotype from homeostatic state to interferon responsive microglia better than E4, which supports our finding in Fig.4.
- b. From comparison with the phagocytic/non-phagocytic signature genes from Krasemann et al.¹⁴, we identified a similar pattern of overlapping genes with Fig.4 g-h, such as increased expression of *Apoe*, *Axl* and decreased expression of *P2ry12* and *Tmem119*. Overall, these genes were commonly affected in A β E3 and A β E4 injected mice. We speculate that there is commonality between the effect of E3 and E4 native lipoproteins on phagocytic transcriptomic profile.
- c. The study by Zhao et al.¹⁵ examines the transcriptomic of total brain tissue of mice expressing human APOE3 and E4 isoforms. As such, it is different from the transcriptomes of WT or Trem2^{ko} mice expressing mouse *Apoe*. However, in a recently published study¹⁶, we used a similar approach and compared the transcriptome of our APOE3 and E4 mice to the results from Zhao et. al. When comparing the genes in our study with Zhao et al., we found that the commonality of genes was quite low. We speculate these differences are due to differences between tissue and sorted microglia as well as not assessing microglia response to stimuli (acute or chronic neurodegeneration).
- d. In our reply to Reviewer 2 q #7, we discussed the study by Lin et al.¹⁰ and noted that in terms of the functionality, the authors made a similar conclusion as us, namely “*APOE4 microglia-like cells exhibited altered morphologies, which correlated with reduced A β phagocytosis*”. In terms of transcriptomic analysis, this study is different from ours as it examines the transcriptome of iPSC microglia derived from human APOE3 and E4 subjects. In addition, Lin et al. do not assess the transcriptome after *in vitro* treatment or during injury or neurodegeneration.
- e. The study by Marschallinger et al. examines lipid accumulating microglia (LAM) as affected by aging¹⁷. It is also not very relevant to our study because it examines microglia transcriptome in association with aging without examination of injury/A β or effect of APOE lipoproteins.

2. The RNA-seq data in Fig 2 do not support the conclusion that “A β E3 induce stronger microglial response when compared to A β E4” – there are a similar number of DEGs upregulated/downregulated in both groups and a number of processes perturbed.

We stated in the results of the original submission that there were 722 genes upregulated in mice injected with A β E3 and 1072 genes upregulated in A β E4 injected mice (Fig. 2f, Suppl. Table 2). As the reviewer noted the numbers are very similar in A β E3 and A β E4 injected mice. However, we commented that there is a substantial

difference between the genes and biological processes upregulated by A β E3 vs A β E4 injections. As shown on Fig. 2g, upregulated in A β E3 vs A β E4 microglia were many classical microglia-specific genes (*Tmem119* and *Trem2*) as well as genes and biological processes associated with microglia activation such as cytokines and cathepsins. Furthermore, biological processes upregulated by A β E3 injections were such as *GO:0002376~immune system process*, *GO:0031663~lipopolysaccharide-mediated signaling pathway* that are more connected to active microglia phenotype. In contrast, the genes upregulated in A β E4 injected mice and biological process associated with them were less specific to microglia and its activation (such as *GO:0007155~cell adhesion*; *GO:0030198~extracellular matrix organization*; *GO:0001525~angiogenesis*). We believe that these data support a conclusion that “A β E3 induce stronger microglial response when compared to A β E4” and we added additional sentence to the results.

3. The FACS plots Fig3c do not visually look to be different between E3 and E4, except that E4 look to have slightly higher fluorescence intensity for Abeta-555 (same seems to be true in Fig S3b-d). Could these be represented as a contour plot to make differences more visible to the reader and could the authors please calculate the MFI of Abeta-555 in dual positive cells? Is it possible that more E3 cells take up Abeta but each individual E4 cell takes up more Abeta? Or that phagocytic flux is reduced in E4 and thus more Abeta accumulates? Could the authors please discuss their findings in the context of Muth et al 2019 Front Cell Neurosci, who found that Apoe4 promotes microglial cell motility and upregulates phagocytosis of apoptotic cells, although also found reduced capacity for A β uptake. The authors work supports that of Konttinen et al 2019 Stem Cell reports, who also found reduced migration and phagocytosis, but enhanced cytokine production in APOE4 iP3 derived microglia like cells, and could be discussed.

We changed Flow cytometry plots according to the reviewer critique. Now we present only the single+ and dual+ populations of microglia visualized with a zebra plot (Fig. 3b).

In terms of motility, we tested this in vivo (Fig. 2i-j) and in real time by two-photon (Fig. 3d-e). In the first experiment we used WT mice treated with A β E3 or A β E4 and found that microglia processes are more oriented towards A β injection site in mice treated with A β E3. In terms of real-time two photon imaging we used Cx3cr1^{GFP} mice injected with A β E3 or A β E4 (Fig. 3d-e). The microglia in mice injected with A β was faster reaching and surrounding injection site indicative of better coverage of the infused A β compared to A β E4.

Regarding the study by Muth et al.¹⁸, we believe that there are several important differences compared to ours that could make the comparison difficult. First, we used non-transformed and non-transfected primary microglial cells. In contrast, Muth et al.¹⁸ used transformed cells lines (N9 and Neuro2a) that were additionally transfected with APOE2 or APOE4 constructs. Second, Muth et al. tested the uptake of A β alone in contrast to our study where native APOE lipoproteins were added to A β and as shown it enhanced A β uptake. As shown on Suppl. Fig. 4 in the revised manuscript, the uptake of A β alone is lower compared to when APOE3, E4, or E3^{het} native lipoproteins are added. Furthermore, our study tested the affinity of WT mouse microglia towards A β pre-incubated with native E3 or E4 particles. It is fundamentally different from the Muth et al. study which examined the motility of microglia expressing APOE4 towards A β alone.

We agree with the reviewer that our study support the conclusions made by Konttinen et al.¹⁹ that APOE4 microglia demonstrate a decreased phagocytosis/uptake compared to E3. Our study is different from Konttinen et al.¹⁹ in that it looks at APP mice that are not expressing human APOE isoforms.

4. The authors suggest that a rapid response by E3 microglia to isolate A β may be a protective mechanism diminishing the spread of A β . This should be relatively straightforward to test using their paradigm and assessing A β spread at a later time point. Similarly, later timepoints could be used to assess the efficiency of plaque coverage by microglia.

In the original submission, we used only WT non-APP transgenic mice that do not form amyloid plaques. In order to respond to the reviewer's comment and assess the efficiency of plaque coverage by microglia we used APP/PSdE9 mice expressing human APOE isoforms (referred to as APP/E3 vs APP/E4 mice) at 4-6 months of age¹⁶ (Suppl. Fig.3 in the resubmission). As shown on Fig. 3 in the response, APP mice expressing APOE3 have a higher microglia coverage vs their APOE4 expressing counterparts (panels a-b). The increased microglia barrier in APP/E3 mice is accompanied by different morphology (panels c-d) and there is a significant negative correlation between the length of microglia branches and amyloid plaques load (panel f). We posit that differences in microenvironment around amyloid plaques e.g. the composition of APOE-containing lipoproteins may affect microglia response to amyloid aggregation in plaque form.

Figure 3 (Supplemental Figure 3): In vivo microglial barrier surrounding amyloid plaques and microglia morphology depends on APOE isoform (Supplemental to Figure 2). We first evaluated whether APOE isoform affects the plaque associated microglia barrier across different levels of amyloid deposition using an in vivo amyloid labeling technique (X04) followed by postmortem staining with IBA1 or ThioS in APP/E3 and APP/E4 mice. (a) Representative confocal images of an amyloid plaque (X04 in blue) with I1 (green). (b) Linear regression analysis demonstrates a significantly strong positive correlation between microglia barrier and A β plaques in the cortex of APP/E3 mice (n=11; p<0.01). There is no significant correlation in APP/E4 mice (n=10; p= 0.1734). (d) Microglia process length is significantly decreased and (e) the number of branches shows a trend towards decrease in APP/E4 mice which confirms the results in our infusion model. Analysis was performed by t test. (f) Nonparametric correlation analysis demonstrates a negative correlation between microglia process length and A β plaques in the cortex. Spearman coefficient $r = -0.90$, $p < 0.01$.

In terms of A β spread in WT mice, early studies from Mathias Jucker group demonstrated that brain injection with synthetic A β does not spread in WT as well as APP mice^{20, 21}. Furthermore, our published studies demonstrated that spreading of injected naturally occurring A β oligomers isolated from APP brains depends on the host mouse²². Thus, we did not expect that the synthetic A β oligomers injected into the brain of WT mice could spread beyond the injection site and did not assess that.

5. Can the authors attempt to functionally define each of the 5 microglial clusters? ie Cluster 1a is almost entirely composed of cells from mice infused with E3 lipoproteins- thus even “homeostatic” microglia in an E4 context are not quite “homeostatic”.

Using an unsupervised clustering method, we show that cells in each cluster share similar, specific gene expression patterns across the experimental groups. For example, the cells from each experimental group in Cluster 1a show the highest expression level of homeostatic marker genes compared to other clusters as shown on the heatmap below. Furthermore, Cluster 1a showed no differentially expressed genes between A β E3 vs A β E4 or Trem2^{ko}-A β E3 vs Trem2^{ko}-A β E4.

The following is the description of the clusters as presented in the result section:

- a) **Cells in both Cluster 1a and 1b** expressed high levels of homeostatic genes such as *Cx3cr1*, *P2ry12*, *Tmem119*, and *Siglech*^{12, 13} compared to clusters 2, 3 and 4 (see heat map above). We designated them as homeostatic microglia clusters essential for maintaining CNS homeostasis.
- b) **Cells in Cluster 1b** expressed intermediate features between homeostatic and activated microglia.

Figure 4 (Suppl. Fig. 9c). Heatmaps showing the gene expression profile of HM and DAM genes in each experimental group in each cluster.

These cells showed a slightly decreased expression of homeostatic genes compared to the baseline in cluster 1 (see the heatmap above). Additionally, cluster 1b showed an increase from the baseline (cluster 1a) of the expression of genes related to microglial activation such as *ApoE*, *Cst7*, *Trem2*, and *Tyrobp*. This suggest that cells in this cluster are in transition from homeostatic to active state and were thus designated as transitioning microglia.

- c) **Cells in Cluster 2** show increased expression of DAM genes¹², such as *Spp1*, *Ccl12*, *Fth1*, and *Lpl* which are associated with neurodegenerative disease and significantly decreased expression of homeostatic genes. Although our data come from acute stimuli, these genes' up-regulation in Cluster 3 suggests similar microglial activation between acute and chronic stimuli^{12, 13}.
- d) **Cells in Cluster 3** demonstrate increased expression of proinflammatory cytokine and chemokine genes, such as *Ccl4*, *Tnf*, and *Il1b*, and thus were designated as inflammatory microglia. Together, the cells in Cluster 3 show the feature of activated microglia.
- e) **Cells in Cluster 4** expressed interferon responsive genes, especially type I interferon related genes such as *Ifit3*, *Irf7*, and *Oasl2* and were designated as interferon response microglia²³.

6. Could the authors please comment on E3 function: is E3 preventing oligomer formation and/or inducing stronger microglia responses or both and which is likely to be more important in the brain?

As we explain in the results, A β incubation with either E3 or E4 native lipoproteins virtually abolished the formation of oligomers as examined by EM and Western blotting. In fact, were able to detect A β oligomers only when A β was incubated alone (no addition of E3 or E4). Thus, in terms of preventing oligomer formation both E3 and E4 native particles seem equally effective at preventing it determined by our methods for detection. Our conclusion is that the E3 native lipoproteins induce a stronger microglial response compared to E4 that does not depend on effects on A β oligomerization.

Minor comments:

1. Line 48- species not spices

The spelling has been corrected.

2. Please clarify whether Fig 1a is from previous or current paper – should be referenced not included if the former?

For the Fig. 1, representing human brain lipidomics from APOE3/3, APPOE3/4 carriers and APOE4/4 we used raw lipidomic data which were a part of Lefterov et al study³. These data were re-analyzed because the focus of Lefterov et al. study³ was APOE2 isoform. Thus, the analysis and the panels shown on Fig. 1 are new to the current manuscript. We added explanation that this is a new analysis.

3. Fig 1e – the text is too small to read, it's not clear what is being shown, what the axes represent, what the grey bars represent and why there are 2 bars for most phospholipids, but 3 bars for SM. Are correlations in all samples or in Apoe3/3 or 4/4 samples only? More detail in the figure legend would help clarify these points

Fig 1c shows a correlation matrix of lipidomics data from both humans and ACM. The amount of lipoprotein is compared across species and correlation is determined. The bottom left portion of the figure depicts scatter plots of lipoproteins between given species, with color of the dots corresponding to *APOE* isoform (purple: E3, green: E4). The upper right portion shows correlation coefficients between combinations of lipoproteins. The axes are for the pmol amount of the lipoproteins. The grey bars are histograms of the amount of the given lipoprotein. The number of bars in the histograms is dependent on the density of lipoprotein data for that given species. Correlations are between species. The size of the text has been increased to improve readability.

4. Fig 1f – please include a legend to show what the axes represent

We labeled x and y axis.

5. Fig 1f-h to clarify, are these analyses performed on APOE3/4 lipoproteins isolated from astrocyte-conditioned media? If so, please state more explicitly in methods/results or figure legend

We clarified that panels f-j represent native lipoproteins isolated from astrocyte conditioned media in the figure-legend.

6. Fig S1b-j please label the x axes.

We labeled x and y axes.

7. Fig S2a – please provide inset images for all panels as the structure of oligomers is difficult to see.

As we explained in the response to q #6, we detected A β oligomers only on A β alone preparations marked by red arrows (Suppl. Fig. 2a). We did not detect A β oligomers in A β E3 or A β E4 preparations. With white arrows were labeled native APOE lipoproteins that are visualized as discs that are the same in E3 and E4.

8. Fig 2 – please ensure that all text sizes are consistent.

We changed the font size on Fig. 2 to be consistent.

9. Fig 2 b,d,I,j, please include all datapoints as dot plots to make the n and data distribution clearer for the reader.

We added all datapoints to the column graphs of Fig. 2.

10. Fig 2 ij it's not clear what is being measured here as the heatmap images do not look similar to the Iba1 staining images below. Please clarify what the upper images and % coverage refers to – does this correspond to the individual microglia below or the region?

The method is explained in in details in the Method section and in figure-legend the use of Imaris module is noted.

Methods:

We utilized the FilamentTracer module (Imaris, version 7.1.1, Bitplane) to trace processes of cortical IBA1 positive microglia and determine process length and number of branch points as indicators of microglial morphology complexity. Using the 3D mapped filament tracings, we oriented all microglia so their central axis was positioned facing the infusion site and generated a heatmap depicting the location of microglia from an experimental group utilizing the ImageJ - heatmap from image stack plugin (National Institutes of Health). The number of overlapping microglial processes increases the pixel saturation metric. The percent of all processes which occupy space in each of the 4 quadrants was used to determine the percent coverage of that quadrant.

11. Fig 2lm the data appear to be from 7 mice per group, was one mouse excluded and why?

It was a typing mistake: the number is 7 per group. No mouse was excluded.

12. Fig 4a –microglia misspelt microglai

The spelling was corrected.

13. Fig 4 –keeping the colour scheme for each of the 8 populations consistent would really help visual interpretation of this data e.g. fig 4a v fig4b

It was corrected

14. Fig 4e – please show the error bars for CPM values for the replicates of gene expression

In the revised figure, the error bars are shown on Fig. 4e.

15. Fig 5f – are differences significant?

We used statistic tests (Chisq.test an Binom.test in R) and found significant differences of cell distribution in clusters 1b and 2 between WT and Trem2^{ko}. Cluster 1b – more cells in Trem2^{ko} (p-val = 1.11×10^{-6}) Cluster2 – more cells in WT (p-val = 2.2×10^{-16})

16. Line 350: “we” should be “as”

It was corrected.

Reviewer 4

In this paper, Fitz et al demonstrate a role for APOE genotype in determining cognitive outcomes and microglial phenotypes following A β infusion into the brain. Using in vivo 2-photon imaging, the authors provide evidence to show that APOE genotype affects microglia mobilization towards A β oligomers. The authors also demonstrate that APOE genotype affects microglial transcriptional programs, then investigate the potential interaction between APOE genotype and TREM2. While these data are certainly valuable, the manuscript contains several major weaknesses. The TREM2 data presented in this manuscript lack novelty, and several conclusions made by the authors require further support. In addition, the different areas of the paper are not well connected to each other. Therefore, as discussed below, I think this paper in the current form has not provided sufficiently significant conceptual advances to warrant its publication in this journal.

Major Comments

1. Different parts of the manuscript are somewhat disconnected from each other. While compelling, the lipidomic data in Figure 1 are not directly relevant to the remainder of the paper. In addition, while Figures 2-4 discuss a potential relationship between APOE isoform and microglial function, Figures 5-7 shift towards discussing the effect of TREM2 on DAM gene expression. As presented, these stories are not well integrated to make coherent conclusions.

The main goal of our study was to examine if the different phospholipid composition of APOE3 and E4 native lipoproteins (shown on Fig. 1) when combined with A β has a differential effect on microglia activation. Our rationale to include Trem2^{ko} mice is supported by the published studies demonstrating the role of lipids in TREM2 activation as well as TREM2-APOE interaction. It has been shown that TREM2 senses a broad array of lipids and among them were several of the phospholipid classes identified by us resulting in its activation²⁴. Several groups have shown that TREM2 binds to APOE^{9, 25, 26}. Yeh et al. showed that binding to APOE is increased by APOE lipidation and facilitates A β uptake by microglia⁹, and lipoproteins such as HDL and LDL are considered TREM2 ligands²⁷. Furthermore, variants associated with increased risk for AD were shown to have a markedly decreased binding to TREM2 ligands such as HDL and LDL²⁷. We reasoned that TREM2 or other microglia receptors could be affected by the APOE isoform dependent level of lipidation resulting in differential A β uptake in APOE3 or APOE4 treated cells. In our response to Reviewer 2, we mentioned the study by Nugent et al.¹¹ that showed that liposomes containing phospholipid classes shown in our study activate downstream TREM2 signaling.

2. Figures 5-6 focus on the role of TREM2 on DAM gene expression. However, it is well accepted in the field that TREM2 is necessary for homeostatic microglia to transition to the DAM state. These data lack novelty and distract from the other more innovative experiments in this manuscript.

We employed a novel *in vivo* model to test the acute response of Trem2^{ko} microglia to the infusions of A β complexed with APOE3 or E4 native lipoproteins. The experiments shown on Fig. 5-6 compare the response of WT and Trem^{ko} microglia to these injections and are a necessary transition to the novel experiments shown on Fig. 7.

3. The authors have not included a control condition (non-infused or mock-infused) in their single-cell sequencing experiments. This is particularly problematic since the authors conclude that Clusters 2, 3, and 4 appear in response to A β infusion. Without a control, it is not possible to identify which of these populations actually result from A β infusion. Previous scRNA-seq studies have shown that DAM (Cluster 2) and interferon-related (Cluster4) populations are also present in the wildtype brain without treatment. In addition, Cluster 3 is characterized by expression of immediate early genes, which can be induced by experimental artifacts resulting from certain methods of cell isolation²⁸(Haimon et al. Nat Immunol. 2018)— based on the presented data alone, it is unclear how different microglial populations change in response to A β -APOE.

We agree with the reviewer and we added non-injected microglia and performed additional analysis. According to the new data analysis including the non-injected sample, we found clusters showing a similar signature with clusters 2, 3 and 4 of the original analysis that were designated as active response clusters (see Fig. 5a in response). As the new analysis demonstrates, these clusters consist primarily of cells from injected samples, suggesting a response to A β infusion (Suppl. Fig.8 in the resubmission). We do agree that the expression of immediate early genes (IEGs) could be increased by experimental artifacts, as stated in the article the reviewer has provided. However, we find it unlikely that sequestered mRNA, the suggested source of the artefactual IEG expression, would produce read counts large enough to significantly alter the overall profile of the samples. Furthermore, Kracht et al. observed expression of IEGs in human fetal microglia which was able to be confirmed *in situ* using IHC, despite their use of a more traditional sorting method²⁹. Because of our careful filtration for quality of single cells and the similarity of our microglia isolation to Kracht et al.²⁹, we reason that it is unlikely for the cluster identified to be artefactual in nature. This is further supported by IEG expression being higher in non-injected microglia (see Fig. 5b in response). Had the data been artefactual due to infusion, we would have expected to see their expression to be higher in the injected samples.

4. The authors correlate changes in microglia morphology and gene expression with the behavioral outcomes observed in Figure 2, suggesting a relationship between microglia and APOE3-mediated cognitive improvement. However, the involvement of microglia in this behavior is unclear. The authors should provide more causal data to substantiate this suggestion.

Figure 5. We added non-injected microglia and performed additional analysis. (a) Sankey plot shows cluster preservation between the data sets. The top 100 marker genes for each cluster were used for generation of the Sankey plot. (b) Heatmap showing expression of the top 50 marker genes for Cluster 5 (Bonferroni-adjusted p-value < 0.01, logFC > 0.1) in non-injected (control) and injected (WT and Trem2^{ko}) groups. Cluster 5 is analogous to Cluster 3 for the injected mice (as shown in the original submission and resubmission). (c) table for % distribution of cells in each experimental group and cluster.

The behavior data clearly show that Aβ alone causes cognitive deficits. Our overall conclusion based on experiments shown on Figures 2, 3 and 4 is that microglia of mice injected with AβE3 react more efficiently than AβE4 to contain Aβ and diminish its harmful effects. We have shown on Fig. 2IJ that the microglia of mice injected with AβE3 extends more projections which are directed toward the injection site compared to the microglia projections of mice injected with AβE4. The results of two-photon live imaging (Fig. 3de) demonstrate that the infusion of Aβ together with E3 lipoproteins in contrast to E4, induces a more rapid response by microglia to contain and isolate Aβ, suggesting a protective mechanism diminishing the spread of Aβ. Finally, Aβ uptake experiments in vitro (Suppl. Fig. 3) and in vivo (Fig. 3) demonstrate that Aβ E3 is better at promoting the uptake. Thus, by restricting the harmful Aβ effects and increasing its elimination microglia safeguard the cells involved in cognitive function. We have added additional explanation in the results and the discussion.

5. Disease-associated microglia are thought to constitute a restricted subset of microglia that upregulate a set of signature genes. However, the authors compare DAM gene expression for microglia from all three Clusters 2, 3, and 4 (Figure 6a-d). It is unclear how such comparisons can be interpreted based on the conventional understanding of DAM. What is the significance of this gene expression pattern in clusters that are dissimilar to DAM?

As the reviewer noted, disease associated microglia (DAM) was initially identified as associated with neurodegeneration¹², but upregulation of some of these genes were reported in other conditions including injury¹³. We found that genes associated with activated microglia were upregulated in clusters 2, 3, and 4 and accompanied by downregulation of the homeostatic microglial genes. We focused on these clusters in order to define the response of Trem2^{ko} microglia to Aβ or injury in the presence of APOE3 or E4 native lipoproteins as this may have a significance for AD. Thus, these clusters are associated with acute Aβ injury response and in the text, we characterized them as representing the active microglia state. Regardless that these clusters contain cells with significant upregulation of DAM genes, we did not categorize them as DAM clusters per se as they encompass other pro-inflammatory genes (e.g. *Tnf*) which were not included in the original DAM list¹². On Fig. 6b-d we show that the number and the proportion of DAM genes is decreased in Trem2^{ko} microglia as another

way to confirm that Trem2^{ko} microglia are slow to respond to injury and delay the transition from homeostatic to an active state. In response to Reviewer 3 critiques #1 we also compared our data to other relevant mouse studies. In the revised variant, in addition to DAM genes shown in Keren-Shaul et al. study¹², we compared our data to other published studies such as Hammond et al.¹³ and Krasemann et al.¹⁴ that are also relevant to ours. We added these comparisons to the discussion in the revised manuscript.

6. Figure 7 does not effectively illustrate the interaction between TREM2 and APOE genotype on microglia gene expression. It is difficult to arrive at the conclusions reached in the text based on this figure alone. It would help to first present the interaction between TREM2 and each isoform separately, before comparing these interactions to each other as shown in Figure 7.

In the revised manuscript, we compare the response of WT vs Trem2^{ko} microglia for each treatment (WT microglia injected with A β E3 to Trem2^{ko} microglia injected A β E3 and the same for A β E4 injection). With this new comparison analysis, we could determine the interaction between TREM2 and APOE isoforms separately and how APOE isoform-dependent interaction with TREM2 modulates the transcriptomes of individual microglia. We identified a higher number of DEGs in Trem2^{ko} vs WT microglia injected with A β E4 compared to A β E3 injection as well as an increased expression of homeostatic microglial genes in Trem2^{ko} microglia injected with A β E4. The largest increase in DEGs was seen in Cluster 4, the interferon response microglia, with very few DEGs in microglia injected with A β E3 compared to A β E4. These results support our conclusion that Trem2 deficiency may further exaggerate E4 lipoprotein insufficiency to activate microglia.

Minor Comments

1. To strengthen the impact of the single-cell sequencing data, the authors should clarify how many replicates were performed for this experiment. The legend for Figure 5 states that 2 mice were used per group, but was the whole sequencing experiment repeated? This repetition is important, since the authors conclude that cluster sizes differ among wild-type and TREM2 knockout microglia.

The sequencing experiment was performed twice. Each experiment consisted of one A β E3, one A β E4, one A β E3/Trem2^{ko}, and one A β E4/Trem2^{ko}, with the difference being one experiment had males and the other females.

2. The header of the first results section could benefit from being reworded. It not immediately clear that the two groups being compared are the human lipids and the native lipoproteins.

The header was changed to “Common APOE dependent phospholipid signature between human AD brains and native APOE lipoproteins” to better reflect this comparison.

3. Figure 1E is difficult to interpret. To aid understanding, the authors might consider including some additional guidance in the legend or some labels on the figure.

The Fig. 1e is now 1c in the resubmission and the figure legend has been updated to better explain what this panel shows. Below is the updated figure legend for Fig. 1e.

(c) Correlation matrix of the 9 lipid classes analyzed in human and lipoprotein lipidomics data showing correlation scatter plots (bottom-left), histograms of lipid amount (diagonal) and correlation coefficients between lipid classes (top-right). Purple denotes APOE ϵ 3/3, green denotes APOE ϵ 4/4. Axes, pmol for each lipid species.

4. It is unclear where the genes listed on Figure 3 are coming from at this point in the paper, since the authors discuss transcriptional data in Figure 4.

We agree with the reviewer and the genes were removed from Fig. 3. As it is important for the hypothesis, we added this panel to Suppl. Fig. 6.

5. The FACS data in Figure 3C do not look representative of the bar plot quantifying these data. The quantification

shows a ~1.5 to 2-fold increase in dual-positive cells in the ApoE3 condition, but visually this is difficult to appreciate.

The representation of the flow cytometry data was updated for the resubmission. We are now showing only the single+ and dual+ populations of microglia visualized with a zebra plot.

6. In the legend for Figure 3D, the authors state that early time points are depicted in green while late timepoints are depicted in red. However, the opposite is stated in the text (line 185).

This was corrected in the main text and figure legends; early timepoints are depicted in red while later timepoints are depicted in green.

7. While meant to be a comparison between A β E3 and A β E4, the data presented in Supplemental Figures 3E and 3F are identical.

It was technical mistake and it was corrected in the revised manuscript.

References

1. Stukas S, Robert J, Wellington CL. High-density lipoproteins and cerebrovascular integrity in Alzheimer's disease. *Cell Metab* 19, 574-591 (2014).
2. Vitali C, Wellington CL, Calabresi L. HDL and cholesterol handling in the brain. *Cardiovasc Res* 103, 405-413 (2014).
3. Lefterov I, *et al.* APOE2 orchestrated differences in transcriptomic and lipidomic profiles of postmortem AD brain. *Alzheimers Res Ther* 11, 113-113 (2019).
4. Li Z, Shue F, Zhao N, Shinohara M, Bu G. APOE2: protective mechanism and therapeutic implications for Alzheimer's disease. *Mol Neurodegener* 15, 63 (2020).
5. Jiang Q, *et al.* ApoE promotes the proteolytic degradation of A β . *Neuron* 58, 681-693 (2008).
6. Fitz NF, *et al.* Liver X receptor agonist treatment ameliorates amyloid pathology and memory deficits caused by high-fat diet in APP23 mice. *The Journal of neuroscience : the official journal of the Society for Neuroscience* 30, 6862-6872 (2010).
7. Lanfranco MF, Ng CA, Rebeck GW. ApoE Lipidation as a Therapeutic Target in Alzheimer's Disease. *International journal of molecular sciences* 21, (2020).
8. Fitz NF, Nam KN, Koldamova R, Lefterov I. Therapeutic targeting of nuclear receptors, liver X and retinoid X receptors, for Alzheimer's disease. *Br J Pharmacol* 176, 3599-3610 (2019).
9. Yeh FL, Wang Y, Tom I, Gonzalez LC, Sheng M. TREM2 Binds to Apolipoproteins, Including APOE and CLU/APOJ, and Thereby Facilitates Uptake of Amyloid-Beta by Microglia. *Neuron* 91, 328-340 (2016).
10. Lin YT, *et al.* APOE4 Causes Widespread Molecular and Cellular Alterations Associated with Alzheimer's Disease Phenotypes in Human iPSC-Derived Brain Cell Types. *Neuron* 98, 1141-1154 e1147 (2018).
11. Nugent AA, *et al.* TREM2 Regulates Microglial Cholesterol Metabolism upon Chronic Phagocytic Challenge. *Neuron* 105, 837-854 e839 (2020).

12. Keren-Shaul H, *et al.* A Unique Microglia Type Associated with Restricting Development of Alzheimer's Disease. *Cell* 169, 1276-1290.e1217 (2017).
13. Hammond TR, *et al.* Single-Cell RNA Sequencing of Microglia throughout the Mouse Lifespan and in the Injured Brain Reveals Complex Cell-State Changes. *Immunity* 50, 253-271 e256 (2019).
14. Krasemann S, *et al.* The TREM2-APOE Pathway Drives the Transcriptional Phenotype of Dysfunctional Microglia in Neurodegenerative Diseases. *Immunity* 47, 566-581 e569 (2017).
15. Zhao N, *et al.* Alzheimer's Risk Factors Age, APOE Genotype, and Sex Drive Distinct Molecular Pathways. *Neuron* 106, 727-742 e726 (2020).
16. Fitz NF, *et al.* Trem2 deficiency differentially affects phenotype and transcriptome of human APOE3 and APOE4 mice. *Mol Neurodegener* 15, 41 (2020).
17. Marschallinger J, *et al.* Lipid-droplet-accumulating microglia represent a dysfunctional and proinflammatory state in the aging brain. *Nat Neurosci* 23, 194-208 (2020).
18. Muth C, Hartmann A, Sepulveda-Falla D, Glatzel M, Krasemann S. Phagocytosis of Apoptotic Cells Is Specifically Upregulated in ApoE4 Expressing Microglia in vitro. *Front Cell Neurosci* 13, 181 (2019).
19. Konttinen H, *et al.* PSEN1DeltaE9, APPswe, and APOE4 Confer Disparate Phenotypes in Human iPSC-Derived Microglia. *Stem Cell Reports* 13, 669-683 (2019).
20. Meyer-Luehmann M, *et al.* Exogenous induction of cerebral beta-amyloidogenesis is governed by agent and host. *Science* 313, 1781-1784 (2006).
21. Walker LC, Schelle J, Jucker M. The Prion-Like Properties of Amyloid-beta Assemblies: Implications for Alzheimer's Disease. *Cold Spring Harb Perspect Med* 6, (2016).
22. Fitz NF, *et al.* Opposing effects of ApoE/Apoa1 double deletion on amyloid- β pathology and cognitive performance in APP mice. *Brain* 138, 3699-3715 (2015).
23. Moore Z, Mobilio F, Walker FR, Taylor JM, Crack PJ. Abrogation of type-I interferon signalling alters the microglial response to Abeta1-42. *Sci Rep* 10, 3153 (2020).
24. Wang Y, *et al.* TREM2 lipid sensing sustains the microglial response in an Alzheimer's disease model. *Cell* 160, 1061-1071 (2015).
25. Atagi Y, *et al.* Apolipoprotein E Is a Ligand for Triggering Receptor Expressed on Myeloid Cells 2 (TREM2). *J Biol Chem* 290, 26043-26050 (2015).
26. Bailey CC, DeVaux LB, Farzan M. The Triggering Receptor Expressed on Myeloid Cells 2 Binds Apolipoprotein E. *J Biol Chem* 290, 26033-26042 (2015).
27. Song W, *et al.* Alzheimer's disease-associated TREM2 variants exhibit either decreased or increased ligand-dependent activation. *Alzheimers Dement* 13, 381-387 (2017).
28. Haimon Z, *et al.* Re-evaluating microglia expression profiles using RiboTag and cell isolation strategies. *Nat Immunol* 19, 636-644 (2018).
29. Kracht L, *et al.* Human fetal microglia acquire homeostatic immune-sensing properties early in development. *Science* 369, 530-537 (2020).

Reviewers' Comments:

Reviewer #2:

Remarks to the Author:

The authors have addressed all my comments and concerns.

Reviewer #3:

Remarks to the Author:

The authors have performed extensive experimental and computational analyses to address the reviewer's concerns. I am very happy to recommend this manuscript for publication in Nature Communications.

Reviewer #4:

Remarks to the Author:

The authors have addressed the vast majority of my concerns, and the current manuscript has been significantly improved. I only have a few minor comments:

(1) This comment regards analyzing DAM genes in Cluster 3 and 4, which are not DAM clusters (Figure 6c and 6d, original Comment # 5). The authors have nicely demonstrated that Cluster 2 corresponds to DAM by expressing much higher levels of DAM signature genes compared to other clusters (Figure 5f). Even though DAM genes are further downregulated in Cluster 3 and 4 in Trem2 KO, it is very hard to interpret its biological significance, given these genes are already low (and hence functionally insignificant) to begin with. To avoid unnecessary distraction from the major points the authors are trying to make, I suggest to move these panels (Figure 6c and 6d) to supplemental data or remove them from the manuscript altogether.

(2) On line 97, it should be mentioned that the samples used were mouse primary astrocytes.

(3) On line 161-163, it says "as shown on Fig. 2g..." but Tmem119, P2ry13 are not shown on the graph. Please either add new bars for the mentioned genes, or edit the text to reflect what is shown. Also, typo for "cathepsins", and again, no cathepsin genes are shown.

(4) On line 203, it should be Fig 3a for design.

Reviewer #4 (Remarks to the Author):

The authors have addressed the vast majority of my concerns, and the current manuscript has been significantly improved. I only have a few minor comments:

(1) This comment regards analyzing DAM genes in Cluster 3 and 4, which are not DAM clusters (Figure 6c and 6d, original Comment # 5). The authors have nicely demonstrated that Cluster 2 corresponds to DAM by expressing much higher levels of DAM signature genes compared to other clusters (Figure 5f). Even though DAM genes are further downregulated in Cluster 3 and 4 in Trem2 KO, it is very hard to interpret its biological significance, given these genes are already low (and hence functionally insignificant) to begin with. To avoid unnecessary distraction from the major points the authors are trying to make, I suggest to move these panels (Figure 6c and 6d) to supplemental data or remove them from the manuscript altogether.

In our previous response to Reviewer 4, we have tried to explain that regardless of the significant upregulation of DAM genes in clusters 2, 3 and 4 we did not categorize them as DAM but as clusters that represent “active microglia state”. Furthermore, in the resubmission we stressed on the significant downregulation of homeostatic genes in these clusters. It should be noted that in different studies, there are significant variations which genes represent the active microglia state even when only applicable to neurodegeneration or response to amyloid. On one side are DAM¹, followed by IRM (injury responsive microglia²) and lately ARM (amyloid responsive microglia³). To add to this confusion is that IRM is also an abbreviation of Interferon Responsive Microglia⁴ and ARM could apply to Activated Response Microglia⁴. In contrast, there is a better agreement on the list of genes representing homeostatic microglia genes.

Since the genes we identified in our study were associated to acute A β injury and most closely related to DAM and homeostatic gene lists of Keren-Shaul et al. ¹; we used such comparisons only as an indicator of active microglia response or lack of it as is the case of Trem2^{ko} cells (Figure 6). The reviewer is referring to Fig. 6, where we are comparing WT (A β E3 + A β E4) to Trem2^{ko} (A β E3 + A β E4) separately for clusters 2 (Fig. 6b), 3 (Fig. 6c) and 4 (Fig. 6d) which we characterize as representing the active microglia state. As we explained in the manuscript, clusters 3 (Fig. 6c) and cluster 4 (Fig. 6d) exhibited the largest number of down regulated genes in Trem2^{ko} vs WT microglia and also showed the highest mismatch between WT and Trem2^{ko} microglia in terms of homeostatic/DAM gene expression. As shown on Fig. 7, cluster 3 (Fig. 7b) and particularly cluster 4 (Fig. 7c), showed a robust APOE isoform-dependent effect exemplified by the higher number of differentially expressed genes between WT and Trem2^{ko} microglia treated with A β E4 which was not observed in A β E3 challenged microglia.

(2) On line 97, it should be mentioned that the samples used were mouse primary astrocytes.

We corrected the text of the results and added that the primary astrocytes are established from mice expressing human APOE3 and APOE4

(3) On line 161-163, it says “as shown on Fig. 2g...” but Tmem119, P2ry13 are not shown on the graph. Please either add new bars for the mentioned genes, or edit the text to reflect what is shown. Also, typo for “cathepsins”, and again, no cathepsin genes are shown.

We added Tmem119 and P2ry13 to the scatter plot (Fig. 2f) and column graph (Fig. 2g) and noted that cathepsins are not shown.

(4) On line 203, it should be Fig 3a for design.

This was corrected.

- 1 Keren-Shaul, H. *et al.* A Unique Microglia Type Associated with Restricting Development of Alzheimer's Disease. *Cell* **169**, 1276-1290.e1217, doi:10.1016/j.cell.2017.05.018 (2017).
- 2 Hammond, T. R. *et al.* Single-Cell RNA Sequencing of Microglia throughout the Mouse Lifespan and in the Injured Brain Reveals Complex Cell-State Changes. *Immunity* **50**, 253-271 e256, doi:10.1016/j.immuni.2018.11.004 (2019).
- 3 Nguyen, A. T. *et al.* APOE and TREM2 regulate amyloid-responsive microglia in Alzheimer's disease. *Acta Neuropathol* **140**, 477-493, doi:10.1007/s00401-020-02200-3 (2020).
- 4 Sala Frigerio, C. *et al.* The Major Risk Factors for Alzheimer's Disease: Age, Sex, and Genes Modulate the Microglia Response to Abeta Plaques. *Cell Rep* **27**, 1293-1306 e1296, doi:10.1016/j.celrep.2019.03.099 (2019).